# Scleraxis-lineage cells are required for tendon homeostasis and their depletion induces an accelerated extracellular matrix aging phenotype

Antonion Korcari[1,2], Anne EC Nichols[1], Mark R Buckley[1,2], Alayna E Loiselle[1,2]*

[1]Center for Musculoskeletal Research, Department of Orthopaedics & Rehabilitation, University of Rochester Medical Center, Rochester, United States; [2]Department of Biomedical Engineering, University of Rochester, Rochester, United States

**Abstract** Aged tendons have disrupted homeostasis, increased injury risk, and impaired healing capacity. Understanding mechanisms of homeostatic disruption is crucial for developing therapeutics to retain tendon health through the lifespan. Here, we developed a novel model of accelerated tendon extracellular matrix (ECM) aging via depletion of *Scleraxis-lineage* cells in young mice (Scx-DTR). Scx-DTR recapitulates many aspects of tendon aging including comparable declines in cellularity, alterations in ECM structure, organization, and composition. Single-cell RNA sequencing demonstrated a conserved decline in tenocytes associated with ECM biosynthesis in aged and Scx-DTR tendons, identifying the requirement for Scleraxis-lineage cells during homeostasis. However, the remaining cells in aged and Scx-DTR tendons demonstrate functional divergence. Aged tenocytes become pro-inflammatory and lose proteostasis. In contrast, tenocytes from Scx-DTR tendons demonstrate enhanced remodeling capacity. Collectively, this study defines Scx-DTR as a novel model of accelerated tendon ECM aging and identifies novel biological intervention points to maintain tendon function through the lifespan.

## Editor's evaluation

This fundamental work advances our understanding of the cellular and molecular changes of the aged tendon. The evidence supporting the conclusion is convincing, using a DTR-based ScxLin cell depletion model along with state-of-art proteomic and scRNA-seq analyses. This paper is of potential interest to scientists and physicians who study the mechanisms of the tendon aging process.

*For correspondence:
alayna_loiselle@urmc.rochester.edu

Competing interest: The authors declare that no competing interests exist.

## Introduction

Tendons are dense connective tissues that transmit muscle-generated forces to bone to enable skeletal movement and joint stability. Recent studies from our group and others have demonstrated the presence of multiple tendon resident cell populations, with Scleraxis-lineage (Scx[Lin]) cells being the predominant population during adult mouse homeostasis (*Best et al., 2021*; *De Micheli et al., 2020*; *Best and Loiselle, 2019*; *Kendal et al., 2020*). Scleraxis, a basic helix-loop-helix transcription factor, is the most well-characterized tendon marker (*Schweitzer et al., 2001*; *Murchison et al., 2007*; *Pryce et al., 2009*) and is required for normal tendon development (*Schweitzer et al., 2001*; *Murchison et al., 2007*). During adulthood, tendon homeostasis is maintained via ongoing turnover of extracellular matrix (ECM) proteins (*Samiric et al., 2004a*; *Samiric et al., 2004b*; *Choi et al., 2020*; *Heinemeier et al., 2016*; *Rees et al., 2009*; *Korcari et al., 2021b*; *Neidlin et al., 2018*). Homeostatic

disruptions can lead to the development of tendon pathologies such as tendinopathy, which are characterized by significant pain and permanent decline in tissue performance (*Millar et al., 2021*). While repetitive loading in sports or occupational contexts disrupt homeostasis, natural aging is also associated with impaired tendon homeostasis (*Pardes et al., 2017*; *Connizzo et al., 2013*; *Peffers et al., 2014*; *Gehwolf et al., 2016*; *Wang et al., 2021*; *Kostrominova and Brooks, 2013*; *Marqueti et al., 2018*; *Korcari et al., 2021a*). Tendinopathy prevalence increases substantially with age (*Riel et al., 2019*), making aging a key risk factor for loss of tendon health. Despite this increased risk, the factors responsible for inducing age-related tendinopathy are not well defined.

We have previously shown that depletion of Scx^Lin cells results in a relatively rapid loss of ECM structural and organizational integrity of the flexor digitorum longus (FDL) tendon (*Best et al., 2021*), suggesting that Scx^Lin cells are required to maintain tendon homeostasis. However, the specific mechanisms by which Scx^Lin cells maintain tendon homeostasis have not been identified. Moreover, while the initial ECM organizational changes that occur with Scx^Lin cell depletion mimic those that occur during natural aging (*Connizzo et al., 2013*; *Gehwolf et al., 2016*; *Dunkman et al., 2013*; *Ippolito et al., 1980*; *Connizzo et al., 2016*), it is unknown whether sustained Scx^Lin cell depletion in young mice may induce an aged tendon phenotype.

In the present study, our primary objective was to identify the underlying changes in the cellular and molecular environment that result from Scx^Lin cell depletion in young adult tendons and determine how these structural and compositional shifts compare to those observed during natural aging. Here, we define the mechanisms that are conserved between Scx^Lin depletion and natural aging to establish the key drivers that underpin the initiation of tendon degeneration and identify novel intervention points to maintain tendon health through the lifespan. Conversely, by defining the divergent molecular programming shifts in the tendon cells that remain during natural aging and inducible depletion of Scx^Lin tenocytes in young animals, we have identified cell populations and processes that may dictate differences in tendon healing capacity.

## Results

### Scleraxis-lineage cell depletion in young adults disrupts tendon homeostasis and mimics the cell density and ECM structural alterations of tendon aging

To determine the long-term depletion efficiency of this Scx-DTR (DTR) model, we quantified cell density in DTR mice at 3-, 6-, and 9 months (3, 6, 9 M) post-depletion, and compared to age-matched DT-treated wildtype (WT) littermates (*Figure 1A*). At 3 M and 6 M post-depletion, there was a respective 57.44% (p<0.0001) and 56.21% (p<0.0001) reduction in total tendon cell density in DTR tendons relative to the WT littermates (*Figure 1B and C*). By 9 M post-depletion (12 M of age), there was an age-related decrease in tendon cell density in WT such that no significant differences in cell density were observed between DTR and WT (p>0.05) (*Figure 1B and C*).

Based on this decline in cellularity from 9 to 12 M in WT, we further tracked changes in tendon cell density from 10 to 31 months of age in C57BL/6 J mice (*Figure 1D*). Consistent with the progressive decline in cellularity in WT tendons, cellularity was further decreased at 10 M in C57Bl/6 J tendons, with a 31.62% (p<0.0001) decline in cell density compared to 6 M old WT, and a 32.06% decrease (p<0.0001) compared to 9 M old WT tendons (*Figure 1D and E*). At 13 and 31 M, C57Bl/6 J FDL tendons showed a 43.19% (p<0.001) and a 44.42% (p<0.001) decrease in total cell density compared to 10 M old C57Bl/6 J FDL tendons, respectively (*Figure 1D and E*). Intriguingly, 6 M old DTR tendons (3 M post-depletion) exhibited a cell density almost identical to 13 M and 31 M old C57Bl/6 J tendons (p>0.05) (*Figure 1D and E*), suggesting that with Scleraxis-lineage depletion, young tendons exhibit the same cell density as old (12 M) and geriatric (31 M) tendons (*Figure 1D and E*).

We then investigated the impact of sustained Scleraxis-lineage cell loss on long-term maintenance of collagen ECM organization. Quantification of collagen dispersion via SHG imaging demonstrated significant increases in dispersion (loss of organization) at 3 M (39.82%, p<0.05), 6 M (41.04%, p<0.01), and 9 M (24.53%, p<0.05) post-DT, relative to WT littermates (*Figure 1F and G*), indicative of impairments in tissue structure and organization.

We then determined how the structure and organization of the tendon shifts with natural aging, and whether there may be conserved structural changes with both Scleraxis-lineage depletion and

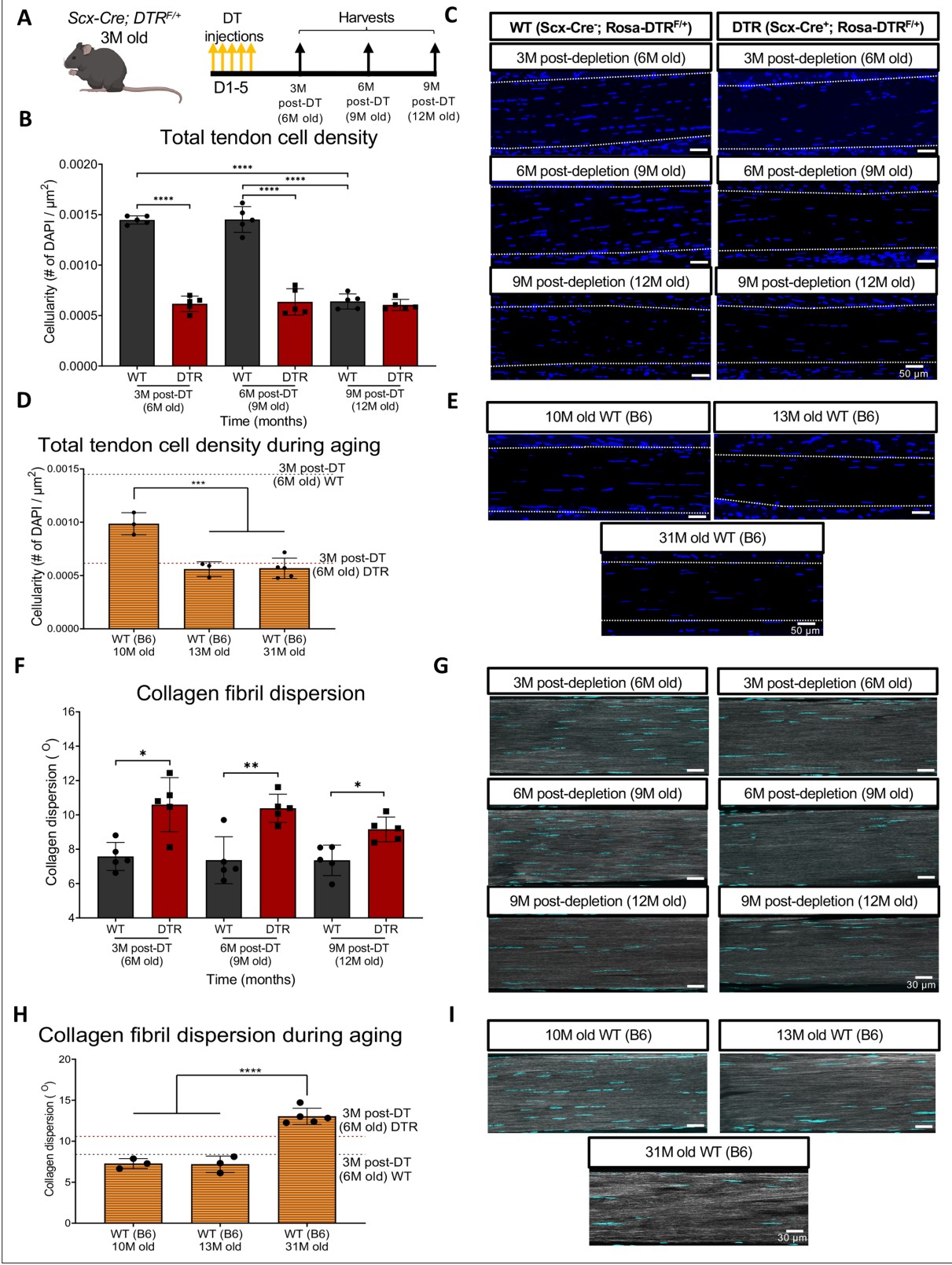

**Figure 1.** Depletion of Scleraxis-lineage cell during long-term homeostasis significantly disrupts tendon structure and mechanical properties. (**A**) 3 M old Scx-Cre+; Rosa- DTR$^{F/+}$ (DTR) and Scx-Cre-; Rosa-DTR$^{F/+}$ (WT) mice received five hind paw injections of DT and were harvested at 3, 6, and 9 M post-depletion. (**B**) Quantification of total tendon cell (DAPI) density from injected WT and DTR hind paws at 3-, 6-, and 9 months post-depletion (6, 9, and 12 months old of age, respectively) within the tendon. (**C**) Representative sections from (**B**). (**D**) Quantification of total tendon cell (DAPI) density from

*Figure 1 continued on next page*

*Figure 1 continued*

C57BL/6 J hind paws at 10, 13, and 31 months old within the tendon. (**E**) Representative sections from (**D**). (**F**) Quantification of collagen fibril dispersion in WT and DTR samples at 3-, 6-, and 9 months post-depletion (6, 9, and 12 months old, respectively) and (**G**) representative collagen fibril morphology captured via SHG. (**H**) Quantification of collagen fibril dispersion from C57BL/6 J hind paws at 10, 13, and 31 months old and (**I**) representative collagen fibril morphology via SHG. N=3–5 per genotype. Error bars indicate mean ± standard deviation.

aging. Geriatric 31 M old C57Bl/6 J tendons exhibited a significant increase in collagen fibril dispersion compared to 10 M (79.3%; p<0.0001) and 13 M (81.47%, p<0.0001) C57Bl/6 J tendons (*Figure 1H1*), suggesting aging-induced impairments in tendon structure and organization. No differences in dispersion were observed in 3 M post-DT (6 M old) WT tendons compared to 10 M C57BL/6 J (p>0.05), and 13 M C57BL/6 J (p>0.05) (*Figure 1H1*). In contrast, 3 M post-DT (6 M old) DTR tendons showed a 45.8% (p<0.01) and a 47.63% (p<0.01) increase in collagen dispersion compared to 10 M and 13 M C57BL/6 J tendons, respectively (*Figure 1H1*), suggesting that young DTR tendons recapitulate several aspects of age-related deficits in tissue organization. However, geriatric 31 M old C57BL/6 J tendons still had a significantly higher collagen fibril dispersion (+23.02%, p<0.05) compared to 3 M post-DT (6 M old) DTR tendons (*Figure 1H1*), potentially suggesting further degeneration during advanced aging.

## Scleraxis-lineage cells maintain tendon homeostasis by regulating the synthesis of high turnover rate ECM proteins

We next sought to identify the specific biological mechanisms that accompany, and potentially underpin these structural changes by characterizing the proteome of DTR and WT tendons at 3 M post-depletion (6 M old), as well as the proteome of old vs young WT tendons (12 M and 6 M old). At 3 M post-depletion, 30 proteins were significantly different between the DTR and WT groups (*Figure 2A*). ECM proteins were the most decreased category in DTR tendons, accounting for 26.7% of the total proteins. (*Figure 2B*). In addition, cytoskeletal proteins, transporters, and scaffold/adaptor proteins were decreased in DTR, though these decreases in inter- and intracellular proteins are likely due to a ~60% reduction in cellularity of the DTR samples. Functional enrichment analysis of all the downregulated proteins demonstrated that ECM-related molecular functions and cellular components were significantly impaired with depletion (*Figure 2C*). Eleven ECM proteins were decreased in DTR tendons (*Figure 2D*), and they were classified as proteoglycans (PG) (36.4%), glycoproteins (GP) (36.4%), ECM regulators (18.2%), and ECM-affiliated proteins (9.1%) (*Figure 2E*). Strikingly, 72.8% of these ECM proteins were high turnover rate glycoproteins and proteoglycans including CHAD, COCH, and KERA (*Samiric et al., 2004a*; *Choi et al., 2020*; *Heinemeier et al., 2016*; *Tam et al., 2020*). Similar findings were also identified at the 9 M post-depletion timepoint (12 M old) (*Figure 2— figure supplement 1A–F*).

With natural aging, 25 proteins were significantly different between the aged (12 M) and young (6 M) WT tendons (*Figure 2F*). Classification of all downregulated proteins in 12 M tendons determined that 28.6% were ECM proteins (*Figure 2G*) and functional enrichment analysis also showed that ECM-related molecular functions and cellular components were significantly impaired (*Figure 2H*). A total of nine decreased ECM proteins were identified in 12 M WT tendons, relative to 6 M WT (*Figure 2I*), and classified as proteoglycans (22.2%), glycoproteins (22.2%), ECM regulators (33.3%), and collagens (22.2%) (*Figure 2J*). The majority of the above ECM molecules have a high turnover rate, similar to the findings for the 3 M post-depletion timepoints (*Figure 2D and E*). Finally, the proteomes of aged (12 M old) WT tendons vs young (6 M old) DTR were nearly identical (*Figure 2— figure supplement 2A, B*). Collectively, these data further support the hypothesis that Scleraxis-lineage cell depletion in young tendons mimic impairments in ECM-related biological mechanisms that occur during natural aging.

String DB found that in DTR tendons there are two separate groups of downregulated proteins (*Figure 2K*), suggesting two separate mechanisms of ECM-degradation with Scleraxis-lineage depletion. With aging, three separate groups of proteins were downregulated (*Figure 2L*). Interestingly, proteins CHAD, ACAN, and KERA, as well as COCH and NEFL were identified also in the Scleraxis-lineage depleted vs WT (*Figure 2K*), suggesting that these two mechanisms are shared with both natural aging and depletion. Based on these analyses, we identified four ECM molecules (COCH, CHAD, KERA, and ACAN) that were consistently downregulated with both natural aging

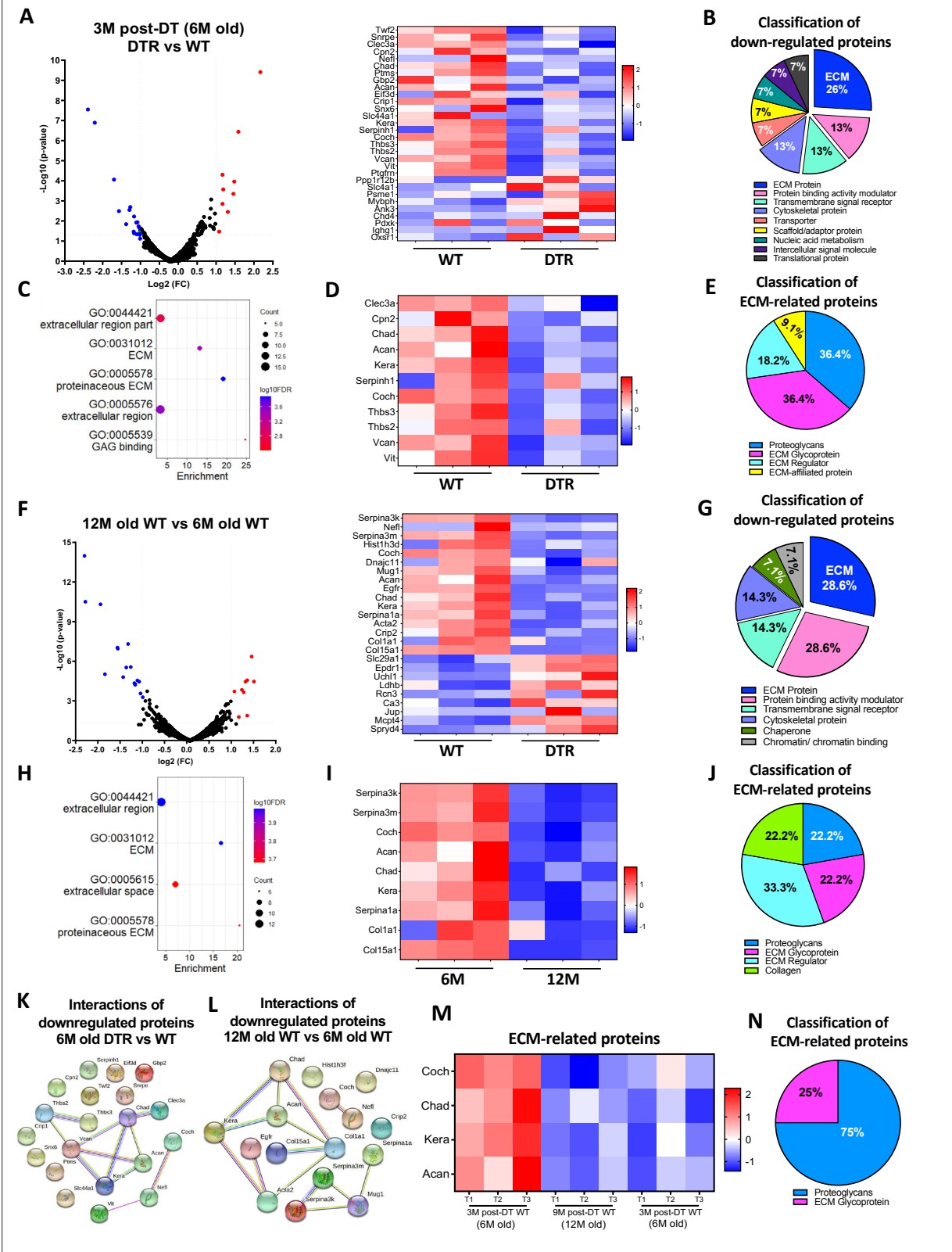

**Figure 2.** Scleraxis-lineage cells maintain FDL tendon homeostasis by regulating the synthesis of high turnover rate ECM proteins. (**A**) Volcano plot and heatmap visualizing the significantly different protein abundances between DTR and WT groups at 3 M post-depletion. (**B**) Classification of all downregulated proteins between the DTR and WT FDL tendons at 3 M post-depletion. (**C**) Functional enrichment analysis of cellular components and molecular functions of all downregulated proteins between the DTR and WT FDL tendons at 3 M post-depletion. (**D**) Heatmap of all differentially

*Figure 2 continued on next page*

*Figure 2 continued*

abundant ECM-related proteins between the DTR and WT FDL tendons at 3 M post-depletion. (**E**) Classification of all ECM-related downregulated proteins between the DTR and WT FDL tendons at 3 M post-depletion. (**F**) Volcano plot and heatmap visualizing the significantly different protein abundances between 12 M and 6 M WT FDL tendons. (**G**) Classification of all downregulated proteins between 12 M and 6 M WT FDL tendons. (**H**) Functional enrichment analysis of all downregulated proteins between 12 M and 6 M WT FDL tendons. (**I**) Heatmap of all differentially abundant ECM-related proteins between 12 M and 6 M WT FDL tendons. (**J**) Classification of all ECM-related downregulated proteins between 12 M and 6 M WT FDL tendons. (**K**) Protein-protein interaction of all the downregulated proteins between DTR and WT groups at 3 M post-depletion. (**L**) Protein-protein interaction of all the downregulated proteins between 12 M and 6 M WT FDL tendons. (**M**) Heatmap with 4 ECM-related proteins that were decreased in similar rates with both natural aging and Scleraxis-lineage cell depletion. (**N**) Classification of the ECM-related proteins from (**M**).

The online version of this article includes the following figure supplement(s) for figure 2:

**Figure supplement 1.** Scleraxis-lineage cells maintain long-term tendon homeostasis via synthesis of high turnover rate glycoproteins and proteoglycans.

**Figure supplement 2.** The proteome between 12 M old WT and young (6 M old) DTR tendons is almost identical with no differences in high turnover rate glycoproteins and proteoglycans.

**Figure supplement 3.** Significant decrease of *Coch* ⁺and *Chad* ⁺ cells with Scleraxis-lineage cell depletion and natural aging.

**Figure supplement 4.** Depletion of Scleraxis-lineage cells significantly impairs tendon structural and material properties.

and Scleraxis-lineage cell depletion (*Figure 2M*), and demonstrated a decrease in the proportion of Coch + and Chad + cells in 6 M old DTR and 21 M old B6 tendons using single-cell RNA sequencing (*Figure 2—figure supplement 3*) Classification showed that 75% are proteoglycans and 25% are glycoproteins (*Figure 2N*). Considering the substantial changes in ECM organization and composition, we also assessed the impact of Scleraxis-lineage cell depletion and natural aging on tendon mechanical integrity (*Figure 2—figure supplement 4A–D*) and while there were no major changes in CSA (*Figure 2—figure supplement 4B*) and stiffness (*Figure 2—figure supplement 4C*), there was a significant decrease of tendon elastic modulus with both depletion and natural aging (*Figure 2—figure supplement 4D*).

Single-cell RNA sequencing demonstrates similar loss of ECM biosynthetic tenocytes as well as differential retention of ECM organizational and age-impaired tenocytes in DTR vs aged tendons.

Given that Scleraxis-lineage cell depletion in young animals induced similar declines in cell density and ECM structure-function, as well as overlapping alterations in ECM composition compared to aged WT tendons, we next asked whether the composition of the cellular environment was conserved in the context of depletion vs. natural aging. Therefore, we performed scRNAseq in 6 M old WT and DTR tendons (3 M post-DT) and 21 M old C57BL/6 J (B6) tendons. 21 M old B6 tendons were used as the 'aged' group as we have previously established that by this age there are age-related impairments in tissue homeostasis and tendon healing response (*Ackerman et al., 2017*).

In the integrated dataset, seven cell types with distinct transcriptomic signatures were identified, including tendon fibroblasts (tenocytes), epitenon cells, endothelial, nerve, and muscle cells, macrophages, and T cells (*Figure 3A and B*; *Figure 3—figure supplement 1*). Annotation of the epitenon cell cluster is based on additional work from our laboratory that will be published in a separate report (*Figure 3—figure supplement 1*). Considering that Scleraxis-lineage cells are tissue-resident fibroblasts (tenocytes), and these are the cells that are subject to depletion, our primary analysis focused on the impact of depletion and aging on tenocytes. We subset and re-clustered the tenocyte population (*Figure 3C and D*) from the integrated dataset (*Figure 3A*). Consistent with cellularity quantification (*Figure 1*), 6 M DTR and 21 M B6 tendons had a decrease in the total number of tenocytes captured compared to 6 M WT (*Figure 3D and E*). To decipher the biological functions of the tenocytes and assess any potential intrinsic programmatic shifts resulting from depletion or natural aging, we investigated their unique transcriptomic profiles by identifying the differentially expressed genes (DEG) between the 6 M WT, 6 M DTR, and 21 M B6 groups (*Figure 3F*), and performing functional enrichment analysis (FEA) based on their top 100 DEG (*Figure 3G–I*).

Tenocytes in the 6 M WT group express genes related primarily to ECM biosynthetic process and secondary to ECM organization such as *Col1a1, Col1a2, Tnxb, Coch, Col11a2, and Fbln1*. Based on FEA, 6 M WT tenocytes are broadly characterized as ECM biosynthetic ('translation', 'macromolecule biosynthetic process', 'collagen metabolic process') and ECM organizational ('ECM organization', 'homeostatic process', 'collagen fibril organization')(*Figure 3F and G*). Consistent with our proteomics findings, broadly, both DTR and aged tenocytes exhibited a decrease in processes related

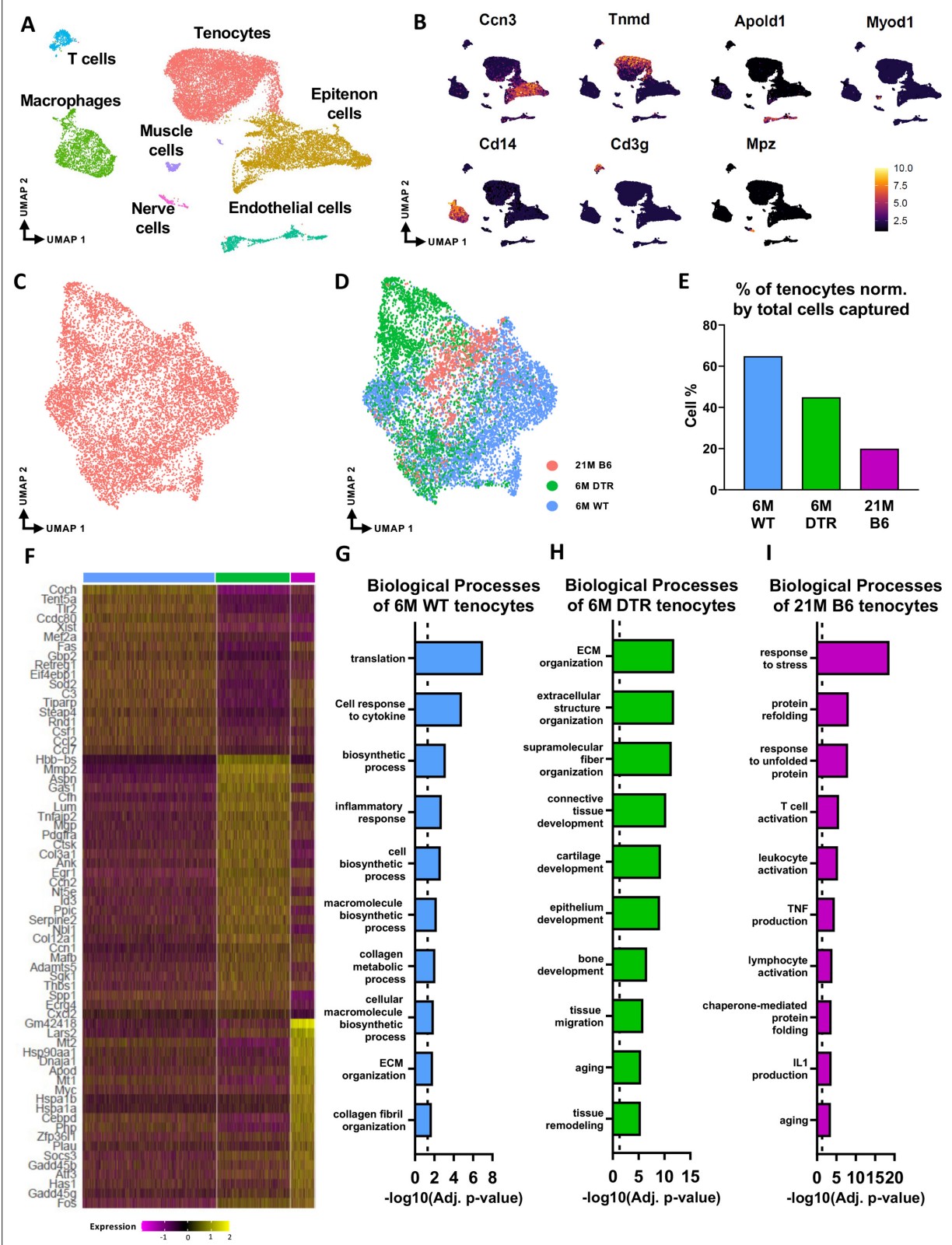

**Figure 3.** scRNAseq demonstrates broad cellular heterogeneity and intrinsic programmatic skewing of tenocytes with depletion and natural aging. (**A**) UMAP dimensionality reduction revealed seven broad and distinct cell populations on clustering based on unbiased differential gene expression of the integrated dataset. (**B**) Annotation marker for each of the seven cell populations identified in the FDL tendons. (**C**) Re-clustering of the tenocytes cell population from (**A**). (**D**) UMAP plot of tenocytes in the integrated data colored based on respective group (pink: 21 M old B6, green: DTR 6 M old,

*Figure 3 continued on next page*

Figure 3 continued

blue: WT 6 M old). (**E**) Quantification of the percentage of tenocytes normalized by the total cells captured per condition. (**F**) Heatmap with the top 40 significantly expressed genes in tenocytes per condition. (**G**) Significantly upregulated biological processes of tenocytes in the 6 M WT (**G**), 6 M DTR (**H**), and 21 M B6 (**I**). Dotted lines in **G**, **H**, and **I** indicate statistical significance of adjusted p-value <0.05 (-log10(adj. p-value = 1.3)).

The online version of this article includes the following figure supplement(s) for figure 3:

**Figure supplement 1.** Annotation of scRNAseq-based identified cell clusters in the integrated data.

**Figure supplement 2.** Tendon resident macrophages exhibit significant age-related intrinsic programmatic shifts.

to ECM biosynthesis such as 'translation', 'biosynthetic process', 'cell and macromolecule biosynthetic process' (*Figure 3F–I*). Interestingly, tissue resident macrophages from aged (21 M B6 tendons) demonstrate similar shifts in biological processes, relative to 6 M WT cells, including enrichment for functions associated with 'protein refolding', and 'response to unfolded proteins' (*Figure 3—figure supplement 2*). To better understand tenocyte heterogeneity as well as the similar shifts of tenocyte biological processes in DTR and natural aging, we subsetted and re-clustered the tenocyte cluster, which resulted in three distinct subpopulations (*Figure 4A*), and then examined the distribution of these subsets among the different groups (6 M WT, 6 M DTR, 21 M B6) (*Figure 4B–E*). With both depletion and aging, we found that tenocytes 1 were decreased to almost identical levels. Specifically, tenocytes 1 subpopulation exhibited a 75.18% decrease with depletion and a 78.9% decrease with aging (*Figure 4B–E*). FEA analysis demonstrated that tenocytes 1 express biological processes related to ECM biosynthesis ('biosynthetic process', 'collagen biosynthetic' and 'collagen metabolic process') and immune response ('inflammatory' and 'immune response', 'response to IL1', 'T cell activation') (*Figure 4F and G*). Finally, we identified *Gbp2* as a specific marker of tenocytes 1 (*Figure 4J*) and found that with both depletion and natural aging there was a decrease in *Gbp2*$^+$ + compared to the 6 M WT (*Figure 4J*). Staining for GBP2 protein validated our scRNAseq results and showed a 58.66% (*p<0.0001*) and a 44.54% *(p<0.0001)* decrease in *GBP2*$^+$ + with depletion and natural aging relative to the 6 M WT group (*Figure 4K and L*). Taken together, our scRNAseq results demonstrate that the consistent loss of ECM biosynthetic tenocytes 1 in young DTR and aged groups is the main driver for impaired tendon homeostasis.

Besides the consistent decrease of ECM biosynthetic tenocytes with DTR and aging, we also identified divergent retention of the remaining tenocytes between DTR and aged groups (*Figure 3F–I*). Broadly, tenocytes in the 6 M DTR group express genes related ECM organization, development and remodeling such as *Edil3, Mmp2, Col5a1* and *Fn1*. Based on FEA, 6 M DTR tenocytes are characterized as specialized cells for ECM organization and remodeling ('ECM organization', 'extracellular structure organization' 'supramolecular fiber organization', and 'tissue remodeling'). In contrast, tenocytes from 21 M B6 mice express genes related to hallmarks of aging like loss of proteostasis (*Ppp1r15a, Hspa8, Hsp90aa1, Hspa5, Hspb1, Hspa1, Hspa1b,* and *Dnajb9*) and inflammaging (*Il6, Stat3, Tnfaip3, Cxcl12,* and *Bcl3*). Indeed, FEA of 21 M B6 tenocytes demonstrated processes related to 'response to stress', 'response to unfolded protein', 'T cell and leukocyte activation', 'TNF and IL1 production', and 'aging' (*Figure 3F1*). Further analysis showed that in the DTR group there was a retention of tenocytes 2, while this population was decreased by 96.2% in the aged WT tendon relative to the young WT (*Figure 4B–E*). Tenocytes 2 express genes that give rise to ECM proteins such as *Mmp2, Lum, Col1a1, Col3a1, Col11a1, and Mfap5*. FEA demonstrated biological processes such as 'ECM and collagen fibril organization', 'supramolecular fiber organization', 'cell adhesion', and 'wound healing', suggesting that tenocytes 2 exhibit functions related to ECM organization and wound healing (*Figure 4F and H*). Taken together, these divergent retention of tenocytes in DTR and aged tendons might dictate divergent healing responses after injury.

In addition to divergence of tenocyte subpopulations shifts, we also assessed for potential intrinsic programmatic shifts of tenocytes 1 and 2 with DTR and natural aging by comparing the transcriptomic profile of tenocytes 1 with DTR and aging (*Figure 5A–J*), as well as the transcriptomic profile of tenocytes 2 in the 6 M WT vs 6 M DTR groups (*Figure 5K–O*). In DTR compared to WT tendons, tenocytes 1 exhibited a total of 413 DEG (*Figure 5A and B*). FEA revealed that tenocytes 1 in the 6 M WT group exhibited processes such as 'translation', 'macromolecule biosynthetic process', 'structure morphogenesis', while in the 6 M DTR group, tenocytes 1 exhibited processes such as 'cell migration', 'ECM organization', and 'wound healing' (*Figure 5C–E*). In contrast, looking at age-related programmatic

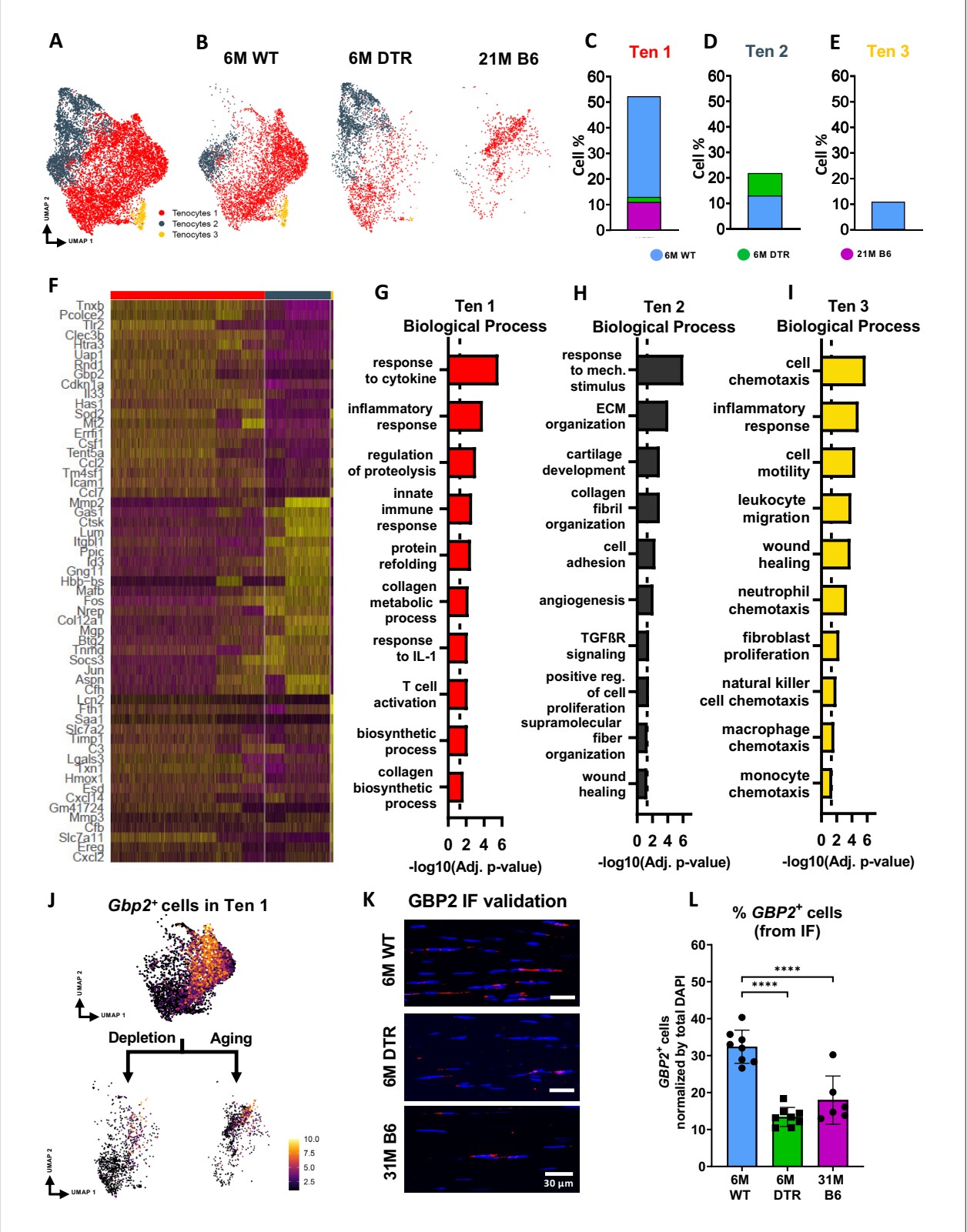

**Figure 4.** DTR and aged tendons lose tenocytes associated with ECM biosynthesis and immune surveillance, while DTR tendons retain tenocytes associated with ECM organization and remodeling. (**A**) UMAP plot of all three tenocytes subpopulations. (**B**) UMAP plot showing the shifts of tenocytes 1–3 with depletion and natural aging. (**C–E**) Quantification of % in tenocytes 1 (**C**), tenocytes 2 (**D**), and tenocytes 3 (**E**), in the 6 M WT, DTR, and 21 M B6 groups (**F**) Heatmap with the top 40 significantly expressed genes per tenocyte subcluster. (**G–I**) Significantly upregulated biological processes of

*Figure 4 continued on next page*

*Figure 4 continued*

tenocytes 1 (**G**), tenocytes 2 (**H**), and tenocytes 3 (**I**), respectively. (**J**) UMAP feature plot of *Gbp2*+ + and their shift with depletion and natural aging, respectively. (**K**) Representative IF of protein validation on the decrease in GBP2 +tenocytes 1 cells with Scleraxis-lineage cell depletion and natural aging. (**L**) Quantification of IF for GBP2 + cell density in 6 M WT, 6 M DTR, and 31 M B6 groups. Data are presented as mean ± standard deviation.

shifts of tenocytes 1 (*Figure 5F–H*), while young WT tenocytes 1 exhibited processes such as 'collagen biosynthesis', 'ECM organization', and 'immune cell chemotaxis' (*Figure 5I*), aged tenocytes 1 exhibited processes related to potential indicators of aging hallmarks such as loss of proteostasis 'response to unfolded protein', 'protein refolding', and 'inflammaging' 'T cell activation', 'immune response' (*Figure 5J*). Regarding programmatic changes of tenocytes 2 in 6 M WT vs DTR (*Figure 5K*), FEA analysis from the DEGs (*Figure 5L and M*) showed that tenocytes 2 in the young WT tendons exhibited biological processes such as '*macromolecule biosynthetic and metabolic processes*', '*collagen metabolic processes*', and '*ECM organization*', while tenocytes 2 in the young DTR group exhibited further elevated as well as new processes related to matrix organization and remodeling such as '*ECM organization*', '*extracellular fibril organization*', and '*ECM assembly*' (*Figure 5N and O*). These divergent intrinsic programmatic shifts in tenocytes between DTR and aged tendons might also dictate the divergent healing responses that are observed in these models.

## Tenocyte autocrine and paracrine signaling is differentially altered with Scleraxis-lineage depletion vs. natural aging

Cellular communication is crucial for maintenance of tissue homeostasis (*Bissonnette et al., 2020*; *Lee and Schwarz, 2020*; *Franklin, 2021*; *Mattes and Scholpp, 2018*; *Park et al., 2017*), and physiological communication patterns are impaired with aging, leading to disruptions tissue health (*López-Otín et al., 2013*; *Robert and Fulop, 2014*). Therefore, we next determined what the predominant tenocyte communication patterns are in young WT tendon, and how these patterns are changed in the contexts of depletion and natural aging.

Young WT tenocytes 1–3 exhibit both autocrine and paracrine signaling (*Figure 6—figure supplement 1A, B*) Tenocytes 1 exhibit high communication strength as both sender and receiver, tenocytes 2 is the predominant sender but exhibits low intercellular communication strength as a receiver and tenocytes 3 exhibits high communication strength as a receiver but low strength as a sender (*Figure 6—figure supplement 1C, D*). Scleraxis-lineage cell depletion resulted in substantial shifts in tenocyte autocrine and paracrine communication (*Figure 6A–F*). Tenocytes 1 exhibited a decrease in overall communication, tenocytes 2 exhibited an increase in autocrine and paracrine signaling to tenocytes 1, but a decrease in paracrine signaling to tenocytes 3. Finally, tenocytes 3 demonstrated a decrease in autocrine and paracrine signaling to tenocytes 1, and an increase in paracrine signaling to tenocytes 2 (*Figure 6B*). Five signaling pathways (CHAD, LIFR, NECTIN, CADM, EPHA) were identified exclusively in the 6 M WT group, while four signaling pathways (IL6, SEMA3, NOTCH, and FGF) were expressed only in the 6 M DTR group (*Figure 6C*). The heatmaps demonstrate the strength of outgoing (*Figure 6D*), and incoming (*Figure 6E*) signaling patterns in tenocytes. We further focused on the THBS signaling pathway since *Thbs2, Thbs3,* and *Comp* were significantly decreased with depletion in both the proteomics and scRNAseq datasets (*Figure 2A and D*; *Figure 6—figure supplement 1*). More specifically, we identified that *Thbs3-Sdc1, Thbs2-Sdc4, Thbs2-Sdc1,* and *Thbs2-Cd47* ligand-receptor interactions were decreased in tenocytes 1–3 with depletion (*Figure 6F*), suggesting that this interaction may be important for maintenance of tendon homeostasis.

In contrast to altered communication in response to DTR, natural aging results in an almost complete loss in tenocyte communication compared to 6 M WT (*Figure 6G–L*). Tenocytes 1 and 2 exhibited a loss of both autocrine and paracrine communication (*Figure 6H*). A majority of the signaling pathways were expressed only in the 6 M WT group, suggesting that with natural aging there is a substantial and widespread loss of tenocyte-tenocyte communication (*Figure 6I–K*). Interestingly, SPP1 and WNT pathways were present only in the aged group in both an outgoing and incoming pattern (*Figure 6J and K*). Finally, we also identified a loss of THBS signaling with natural aging, including *Thbs3-Sdc4, Thbs3-Sdc1, Thbs2-Sdc4, Thbs2-Sdc1,* and *Thbs2-Cd47* (*Figure 6L*), which was very similar to the ligand-receptor interactions lost with depletion (*Figure 6F*). Taken together, aging results in broad decline in tenocyte-tenocyte communication, and future work will define whether decreased and

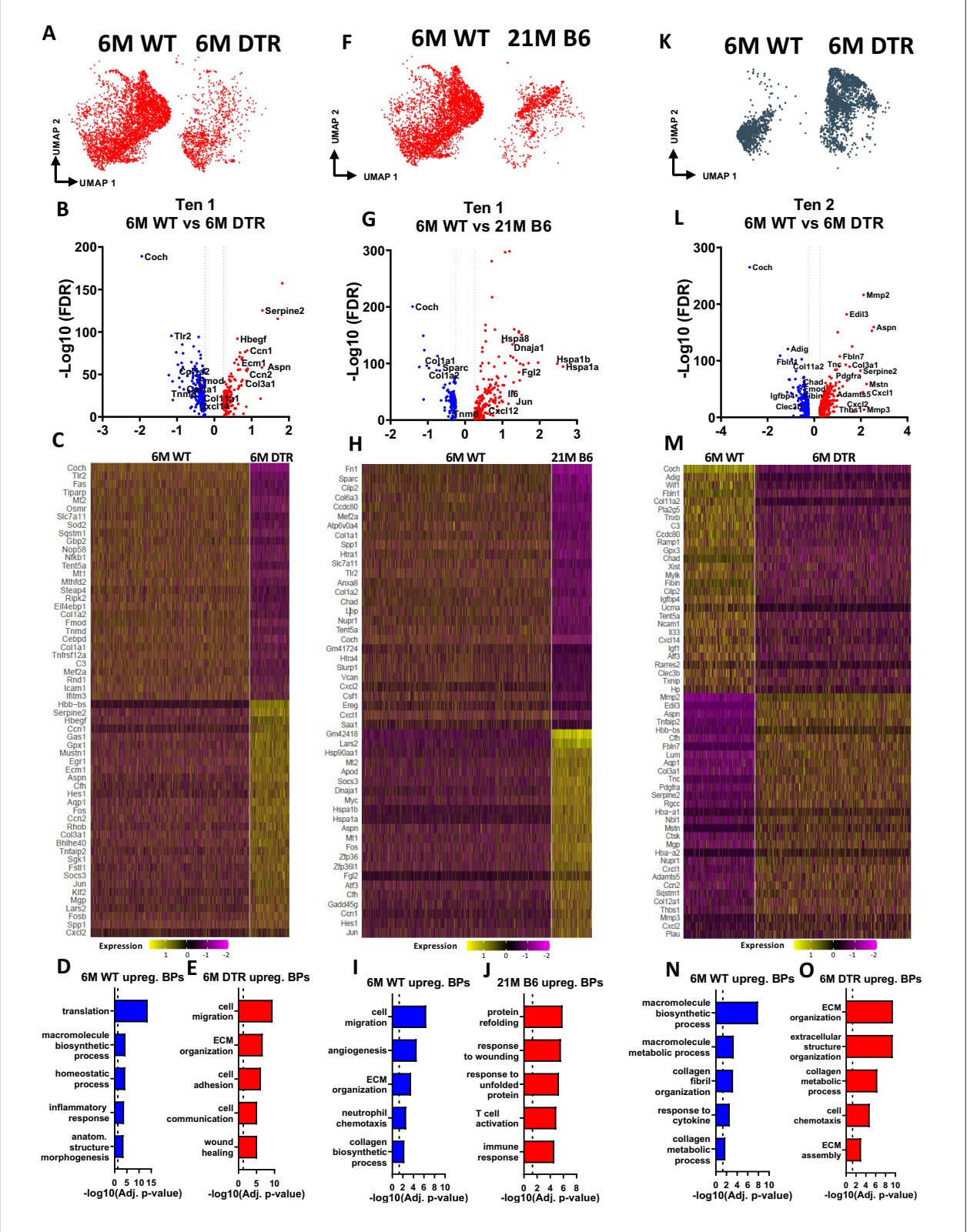

**Figure 5.** Tenocytes 1 and 2 become more ECM organizational in DTR tendons while tenocytes 1 exhibit indications of aging hallmarks such as loss of proteostasis and inflammaging. (**A**) UMAP plot of tenocytes 1 in 6 M WT and DTR groups. (**B**) Volcano plot visualizing all the significantly different genes in tenocytes 1 between 6 M WT vs. 6 M DTR. Blue dots indicate genes that are down-regulated in DTR vs WT, while red indicates genes up-regulated in DTR vs. WT. (**C**) Heatmap of the top 50 differentially expressed genes of tenocytes 1 between 6 M WT and DTR groups. (**D**) Biological processes up

*Figure 5 continued on next page*

*Figure 5 continued*

regulated in 6 M WT tenocytes 1. (**E**) Biological processes upregulated in 6 M DTR tenocytes 1. (**F**) UMAP plot of tenocytes 1 in 6 M WT and 21 M B6 groups. (**G**) Volcano plot visualizing all the significantly different genes of tenocytes 1 between 6 M WT and 21 M B6. Blue dots indicate genes that are down-regulated in 21 M B6 vs. 6 M WT, while red indicates genes up-regulated in 21 M B6 vs. 6 M WT. (**H**) Heatmap of the top 50 differentially expressed genes of tenocytes 1 between 6 M and 21 M B6 groups. (**I**) Biological processes of tenocytes 1 in 6 M WT group. (**J**) Biological processes of tenocytes 1 in 21 M B6 group. (**K**) UMAP plot of tenocytes 2 in 6 M WT and 6 M DTR groups. (**L**) Volcano plot visualizing all the significantly different genes of tenocytes 2 between 6 M WT vs. 6 M DTR. Blue dots indicate genes that are down-regulated in DTR vs WT, while red indicates genes up-regulated in DTR vs. WT. (**M**) Heatmap of the top 50 differentially expressed genes of tenocytes 2 between 6 M WT and 6 M DTR groups. (**N**) Biological processes upregulated in 6 M WT Tenocytes 2. (**O**) Biological processes upregulated in 6 M DTR Tenocytes 2.

altered communication between the remaining tenocytes may drive further degeneration beyond that observed at 21 M B6.

## Discussion

Aging impairs tendon homeostasis, increases the risk of degeneration (*Riel et al., 2019*; *Teunis et al., 2014*) and the frequency of injury, and diminishes healing capacity after injury (*Ackerman et al., 2017*; *Teunis et al., 2014*; *Houshian et al., 1998*). In addition, clinical studies have shown that age-related tendon changes are associated with impaired dynamic stability and increased risk of falling, factors that significantly increase both morbidity and mortality (*Onambele et al., 2006*; *Tomlinson et al., 2021*; *Karamanidis et al., 2008*). In this study, we identified the cell and molecular mechanisms that maintain tendon homeostasis during adulthood and demonstrate how these mechanisms are disrupted with aging. Moreover, we demonstrated that depletion of Scleraxis-lineage cells in the adult tendon mimics age-related impairments in cellular density, ECM structure, composition, and material quality. By comparing these two models, we were able to delineate the mechanisms that regulate tendon homeostasis, independent of the systemic effects of aging, while also establishing the tendon-specific response to aging. More specifically, we have demonstrated that loss of a critical mass of biosynthetic tenocytes, or biosynthetic tenocyte function initiates a degenerative cascade in tendon. This is particularly striking as it demonstrates that it is the decline in cell density and the accompanying loss of cell function itself that drives age-related tendon degeneration, rather than age-related programmatic shifts. Finally, these data provide more granular information on the timeline in which age-associated declines in tendon cell density occur and provide a functional and mechanistic link between declines in tendon cell density and initiation of tendon degeneration. Moreover, we have identified preventing the loss of biosynthetic tenocytes (whether via apoptosis or other mechanisms) as a critical intervention point for maintaining tendon structure-function.

While we have previously shown that this Scleraxis-lineage depletion strategy results in a ~60% decrease in tendon cell density (*Best et al., 2021*), it was unknown whether this was a sustained or transient depletion event that eventually results in re-population of the tendon via compensation by non-depleted populations. Indeed, this model results in sustained cell depletion, thus allowing assessment of the long-term impact of Scleraxis-lineage cell deficiency. It is important to note that this model likely depletes any 'progenitor' cell populations due to the use of non-inducible Scx-Cre, further inhibiting any potential cellular rebounding post-depletion. We observed that Scleraxis-lineage cell depletion resulted in almost identical cellular density with older (12 M old) and geriatric (31 M old) tendons, indicating that with depletion, there is an acceleration of the natural age-related tendon cell loss. An age-related decline in tendon cell density has been extensively documented, with consistent decreases across anatomically distinct tendons and animal models (*Korcari et al., 2023*; *Sugiyama et al., 2019*; *Stanley et al., 2007*; *Yan et al., 2020*; *Freedman et al., 2022*). However, the fact that Scleraxis-lineage depletion in young animals results in a comparable decline in cellularity suggests there may be mechanisms that maintain a minimum level of cellularity and prevents complete cell loss, although these mechanisms remain to be defined. Moreover, the recapitulation of many established hallmarks of tendon aging, including altered ECM structure, organization, and material quality (*Connizzo et al., 2013*; *Gehwolf et al., 2016*; *Dunkman et al., 2013*; *Sugiyama et al., 2019*; *Ippolito et al., 1980*; *Watanabe et al., 1994*; *Watanabe et al., 1997*), further supports the utility of defining the consistent underlying mechanisms between DTR and aging to define previously unappreciated drivers of age-related tendon pathology. Indeed, the combination of transcriptomic and proteomic

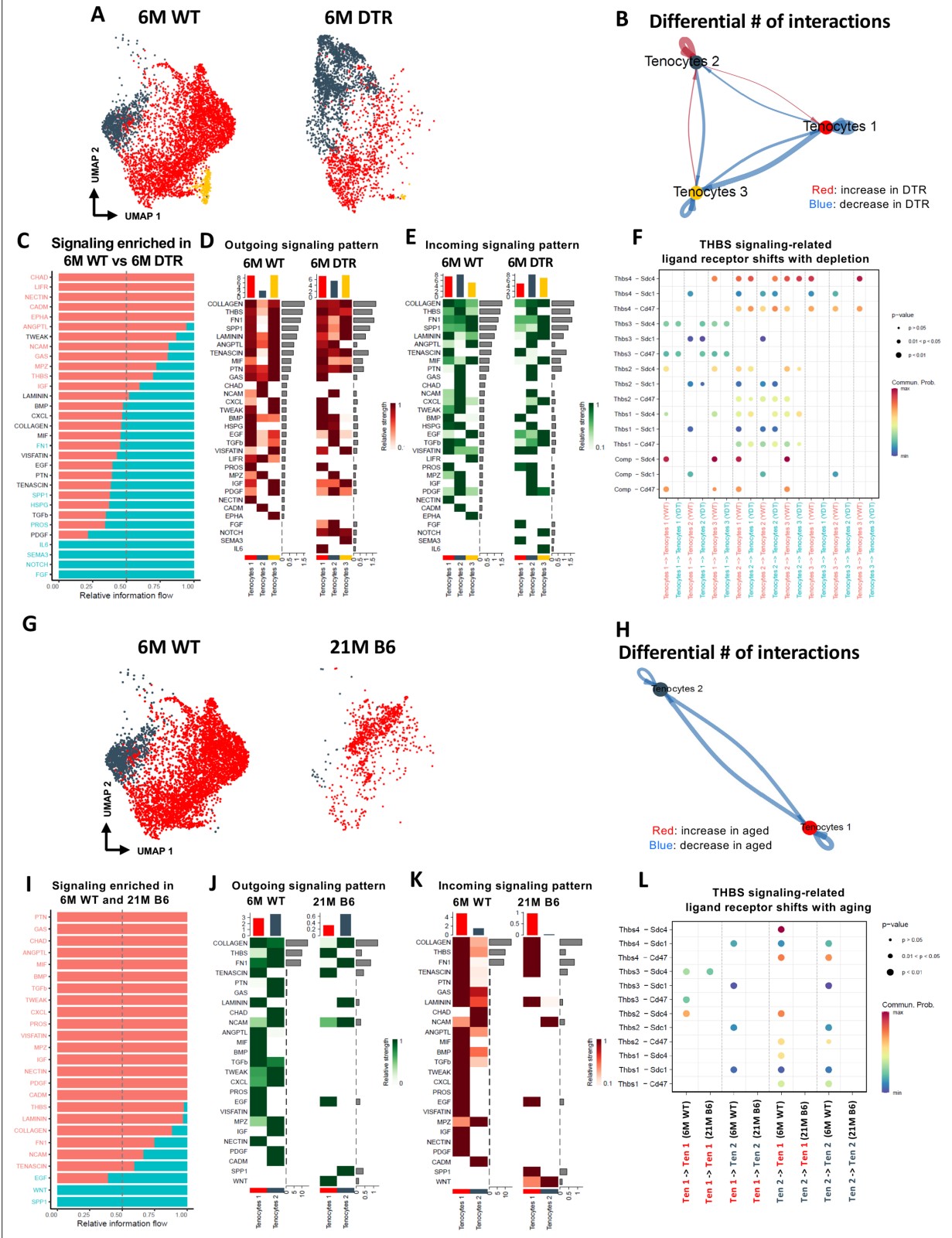

**Figure 6.** Tenocyte-tenocyte communication is impaired with both Scleraxis-lineage cell depletion and natural aging. (**A**) UMAP plot of tenocytes subpopulations in the 6 M WT and DTR groups. (**B**) Differential number of cell-cell interactions of tenocytes 1–3 in the 6 M WT and DTR groups (red color indicates an increase and blue color indicates a decrease in cell communication with Scleraxis-lineage cell depletion relative to WT, respectively; arrow indicates the communication between two different cells and the direction of the arrow indicates the cell that expresses the ligand vs the cell that

*Figure 6 continued on next page*

*Figure 6 continued*

expresses the receptor, where the pointed cell by the arrow is the receiver). (**C**) All the signaling pathways that were identified in the 6 M WT and DTR groups (pathways highlighted with orange color indicate those that were expressed only or higher in the 6 M WT relative to 6 M DTR group; pathways highlighted with green color indicate those that were expressed only or higher in the 6 M DTR relative to 6 M WT group). (**D**) Heatmap with all the outgoing signaling patterns for tenocytes 1–3 for the 6 M WT and DTR groups. (**E**) Heatmap with all incoming signaling patterns for tenocytes 1–3 for the 6 M WT and DTR groups. (**F**) THBS signaling-related ligand-receptor interactions that take place in the 6 M WT and DTR groups. (**G**) UMAP plot of tenocytes subpopulations in the 6 M WT and 21 M B6 groups. (**H**) Differential number of cell-cell interactions of tenocytes 1 and 2 in the 6 M WT and 21 M B6 groups (blue color indicates a decrease in cell communication with natural aging relative to WT); arrow indicates the communication between two different cells and the direction of the arrow indicates the cell that expresses the ligand vs the cell that expresses the receptor, where the pointed cell by the arrow is the cell that expresses the receptor or in other words the receiver (**I**) All the signaling pathways that were identified in the 6 M WT and 21 M B6 groups (pathways highlighted with orange color indicate those that were expressed only or higher in the 6 M WT relative to the 21 M B6 group; pathways highlighted with green color indicate those that were expressed only or higher in the 21 M B6 groups relative to 6 M WT group). (**J**) Heatmap with all the outgoing signaling patterns for tenocytes 1 and 2 for the 6 M WT and 21 M B6 groups. (**K**) Heatmap with all incoming signaling patterns for tenocytes 1 and 2 for the 6 M WT and 21 M B6 groups. (**L**) THBS signaling-related ligand-receptor interactions that take place in the 6 M WT and 21 M B6 groups.

The online version of this article includes the following figure supplement(s) for figure 6:

**Figure supplement 1.** Tenocyte-tenocyte communication in the young adult FDL tendons.

analyses in these models has facilitated identification of the key alterations in ECM composition that are associated with initiation and progression of tendon degeneration. Consistent with declines in high-turnover rate proteoglycans and glycoproteins with aging in other tissues (*Choi et al., 2020*; *Ewald, 2020*; *Ariosa-Morejon et al., 2021*; *McCabe et al., 2020*), these classes of proteins were decreased in both aged and young depleted tendons, demonstrating that Scleraxis-lineage cells are required to directly regulate the synthesis and maintenance of multiple GPs and PGs, which are crucial to maintain tendon structure-function, since their decrease results in significantly impaired collagen fibril organization and biomechanical properties. While other studies have shown through genetic KO models, that high turnover rate GPs and PGs are required to maintain tissue (tendon, bone, cartilage, skin) homeostasis during adulthood (*Watanabe et al., 1994*; *Watanabe et al., 1997*; *Batista et al., 2014*; *Hessle et al., 2014*; *Pöschl et al., 2004*; *Cosgrove et al., 1996*; *Hecht et al., 1995*; *Geng et al., 2008*; *Tiedemann et al., 2013*; *Sakai et al., 2016*; *Dex et al., 2017*; *Barbier et al., 2014*; *Kokenyesi et al., 2004*; *Qabar et al., 1994*), they have been primarily descriptive. In contrast, we demonstrate that Scleraxis-lineage cells are a 'master-regulator' of multiple ECM-related proteins during tendon homeostasis. More specifically, we identified decreased expression of COCH and CHAD protein in the tendon ECM in these models, as well as a consistent decline in Coch + and Chad + cells. Interestingly, COCH expression is observed in the spiral ligament of the inner ear, and *Coch-/-* results in degeneration of this ligament (*Kommareddi et al., 2007*), further supporting the potential importance of this molecule in maintenance of tendon structure.

Consistent with prior scRNAseq studies in mouse and human tendons (*De Micheli et al., 2020*; *Kendal et al., 2020*; *Kaji et al., 2021*), we have identified substantial heterogeneity of the tendon cell environment. We have further built on this by establishing the functional consequences of decreasing the subpopulation that is primarily characterized by ECM biosynthetic functions. Interestingly, age-related declines in ECM biosynthetic function have also been identified in mouse and human skin fibroblasts (*Solé-Boldo et al., 2020*; *Salzer et al., 2018*), thus, future work will focus on establishing and disrupting the mechanisms that drive age-related declines in biosynthetic tenocyte number and function. In addition, we found that Guanylate binding protein 2 (GBP2) was a unique marker for the biosynthetic tenocytes 1, allowing us to track and identify this tenocyte biosynthetic subpopulation. While there are no studies that have assessed the function of GBP2 in tendon, other studies have shown that GBP2 exhibits host-defense and anti-microbial functions (*Tretina et al., 2019*; *Kutsch and Coers, 2021*). Based on the innate immune and defense response BPs of tenocytes 1, in addition to their biosynthetic capacity, it could be speculated that this subpopulation may play a role in the immune defense response of tendons after injury. Since tenocytes 1 decreases with aging, it might drive additional tendon intrinsic immune dysregulation after an injury, further impeding the healing capacity of aged tendons. However, future studies are needed to define the function of GBP2 in tendon cells. As such, GBP2 is exclusively used as a marker of Tenocytes 1 in this study. Finally, our CellChat analyses identified that Thbs2-Sdcn4 signaling was significantly impaired with both DTR and aging, suggesting that this pathway might

be important for maintenance of tendon homeostasis. While this specific pathway has not been extensively studied in tendon, previous work has shown that Sdcn4 is required for muscle development, homeostasis, and healing after injury (*Cornelison et al., 2004*; *Rønning et al., 2020*). As such, future studies can better define the role of this signaling axis in tendon health, as it may represent promising therapeutic target.

In addition to comparable declines in biosynthetic function and subsequent initiation of tendon degeneration in these models, there are several informative areas of divergence. First, the tenocytes that remain in aged tendons demonstrate programmatic skewing, including an increased pro-inflammatory signature and loss of proteostasis. While these shifts are not necessary for homeostatic disruption, as evidenced by comparable disruptions in DTR tendons in the absence of this skewing, it is possible that these changes drive subsequent degeneration at timepoints beyond those included in this study. Consistent with this, 31 M B6 tendons exhibit substantial increases in ECM disorganization, relative to DTR. Moreover, while no enrichment of senescent markers was observed in aged tenocytes, it is possible that the pro-inflammatory signature and loss of proteostasis may be a precursor to subsequent tenocyte senescence, which could be an important driver of additional tendon degeneration. Finally, in addition to the unique programmatic shifts observed in aged tenocytes vs. the young depletion model, aged tendon resident macrophages demonstrate similar shifts in biological processes toward a response to protein unfolding, suggesting this may be a conserved feature of tendon cell natural aging. Another intriguing consideration is how these divergent shifts in tenocyte programs may relate to the divergent healing outcomes observed between these models (*Figure 7*). Consistent with impaired healing capacity in many aged tissues (*Ackerman et al., 2017*; *Browder et al., 2022*; *Sgonc and Gruber, 2013*; *Yamaguchi et al., 2021*), we have previously shown that 22 month-old tendons heal in a mechanically inferior manner, relative to young tendons, and fail to form the provisional ECM needed for successful healing (*Ackerman et al., 2017*). Thus, establishing how these intrinsic programmatic shifts may impair specific aspects of the healing process will be necessary to develop strategies to rescue the aged tendon healing process. In contrast, the cells that remain in DTR tendons display programmatic shifts associated with enhanced remodeling capacity. Moreover, DTR tendons demonstrate improved healing capacity relative to age-matched WT repairs (*Best et al., 2021*). As such, understanding the signals that drive this programmatic shift in DTR, and establishing the mechanisms by which Scleraxis-lineage cell depletion enhances tendon healing holds tremendous potential for tendon tissue engineering and therapeutic strategies. Finally, these data highlight the surprising finding that homeostatic disruptions including decreased cell density and altered ECM composition, quality and organization are not of central relevance to the quality of the tendon healing response, suggesting a de-coupling of the mechanisms that regulate homeostasis and healing capacity.

One limitation of the study is that the *Scx-Cre; DTR $^{F/+}$* targets all Scleraxis-lineage cells, and the Scleraxis-lineage is known to contribute to tissues in addition to tendon (*e.g.* bones and muscle) (*Agarwal et al., 2017*; *Yoshimoto et al., 2017*). However, our initial plans to deplete Scleraxis-lineage cells by utilizing the inducible *Scx-Cre$^{ERT2}$* crossed to the diphtheria toxin A-subunit gene (DTA) mouse model, resulted in insufficient cell depletion. *Scx-Cre$^{ERT2}$*; Rosa-DTR mice could be an alternative to inducibly deplete only adult Scleraxis-lineage cells, although this model requires both TMX and DT administration, and discrepencies in labelling vs. depletion may be observed. In addition, while we clearly demonstrate programmatic skewing in aged (21 M) tendons, relative to young, it is not clear when cells begin to undergo this programmatic shift. Future work to define the temporal nature will the establish the onset of these shifts as an additional important intervention point for maintaining physiological tendon cell function.

In summary, our morphological and multi-omics analyses identified Scleraxis-lineage cell depletion as a novel model of accelerated tendon ECM aging based on comparable changes in ECM organization, composition, and material quality with aged tendons. Moreover, we defined loss of a critical mass of biosynthetic tenocyte function as a conserved initiator of disrupted tendon homeostasis and degeneration in both young and aged animals. Thus, identifying strategies to prevent apoptosis of this population is an important priority for maintaining tendon health through the lifespan. Finally, we have demonstrated divergent programmatic skewing between Scleraxis-lineage cell depletion in young animals, and natural aging, and suggest that these differences may underpin the divergent healing capacity observed in these models. Therefore, it will be necessary to develop different

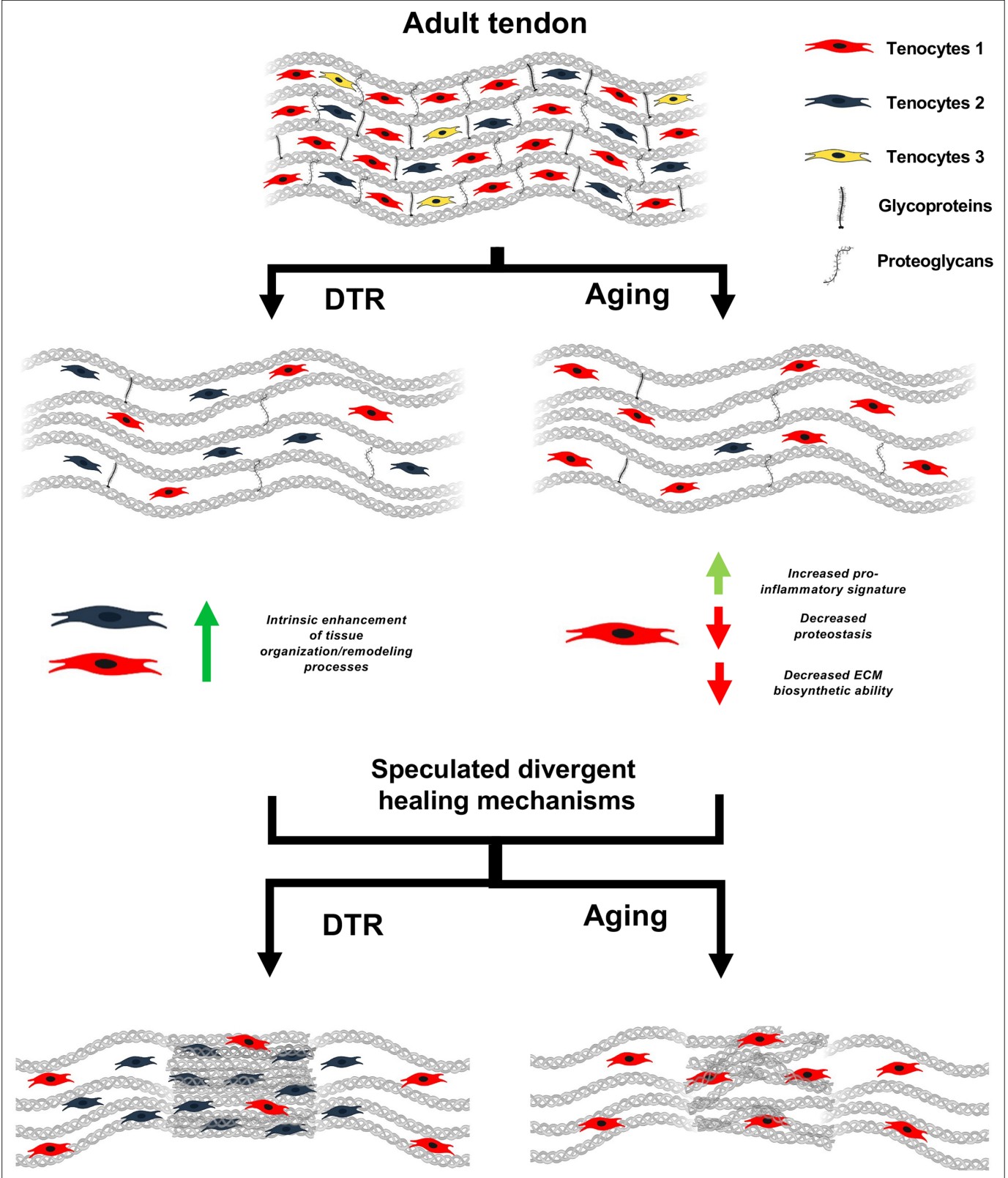

**Figure 7.** Schematic highlighting key findings and proposed models for divergent healing responses in young depleted vs aged tendons. Both young DTR and aged WT tendons have similar decreases in tissue structure, organization, material quality, and total cell density. They also follow the same mechanism of ECM degeneration via a substantial decrease in the number of proteoglycans and glycoproteins with high turnover rate. However, in terms of cell composition shifts with depletion and natural aging, young DTR tendons are comprised mainly of tenocytes specialized in tissue

*Figure 7 continued on next page*

*Figure 7 continued*

organization and remodeling. What is more, young DTR have little to no inflammatory/reactive tenocytes. In contrast, old WT tendons are comprised predominantly of inflammatory and age-related functionally impaired tenocytes and exhibit little to no presence of tissue remodeling tenocytes. We speculate that these significantly different cellular compositions are some of the main causes resulting in divergent healing responses, with young DTR tendons showing improved healing while old WT tendons exhibiting impaired healing response.

strategies to address or prevent these two distinct manifestations of aging in the tendon to preserve both tendon homeostasis and healing capacity.

# Materials and methods

## Key resources table

| Reagent type (species) or resource | Designation | Source or reference | Identifiers | Additional information |
|---|---|---|---|---|
| Genetic reagent (*Mus. musculus*) | *Scx*-Cre | Dr. Ronen Schweitzer | | |
| Genetic reagent (*Mus. musculus*) | C57BL/6-Gt(ROSA)26S or$^{tm1(HBEGF)Awai}$/J (*Rosa-DTR$^{LSL}$*) | Jackson Laboratory | Stock #: 007900 | Referred to as DTR in manuscript RRID:IMSR_JAX:007900 |
| Genetic reagent (*Mus. musculus*) | C57Bl/6 J | Jackson Laboratory | Stock #: 000664 | Referred to as B6 in manuscript RRID:MGI:2159769 |
| Antibody | Anti-rabbit Rhodamine-Red-X (donkey polyclonal) | Jackson ImmunoResearch | Catalog #: 711-296-152 | 1:200 RRID:AB_2340614 |
| Antibody | Anti-GBP2 (rabbit polyclonal) | Proteintech | Catolog #: 11854–1-AP | 1:500 RRID: AB_2109336 |
| Chemical Compound, Drug | Diphtheria Toxin (DT) | Millipore Sigma | Catalog #: D0564-1MG | 20 ng DT / injection |
| Software, algorithm | GraphPad Prism software | GraphPad Prism | https://graphpad.com | Version 9.5 |
| Software, algorithm | OlyVIA software | Olympus | https://www.olympus-lifescience.com/en/support/downloads/ | Version 2.9 |
| Software, algorithm | ImageJ software | ImageJ | http://imagej.nih.gov/ij/ | |
| Software, algorithm | R studio | R Studio | https://www.rstudio.com | |
| Software, algorithm | DAVID Gene Functional Classification Tool | *Huang et al., 2009* | https://david.ncifcrf.gov/ | Version 6.8 |
| Software, algorithm | CellChat | *Jin et al., 2021* | http://www.cellchat.org | Version 1.1.3 |
| Software, algorithm | Seurat R package | *Stuart et al., 2019* | https://www.rdocumentation.org/packages/Seurat/versions/4.1.0 | Version 4.0 |
| Software, algorithm | Discoverer software platform | Thermo Fisher | | Version 2.4 |
| Software, algorithm | PANTHER classification system | *Mi et al., 2021* | http://pantherdb.org/ | |
| Software, algorithm | Search Tool for Retrieval of Interacting Genes/Proteins (STRING) | *Szklarczyk et al., 2019* | | v11.0 |
| Software, algorithm | MatrisomeDB | *Hynes and Naba, 2012*. | https://web.mit.edu/hyneslab/matrisome/ | |

*Continued on next page*

*Continued*

| Reagent type (species) or resource | Designation | Source or reference | Identifiers | Additional information |
|---|---|---|---|---|
| Software, algorithm | *CellMarker* software | *Zhang et al., 2019* | http://bio-bigdata.hrbmu.edu.cn/CellMarker/search.jsp | |
| Other | Gene Expression Omnibus (GEO) | | Accession # GSE214929 | Single-cell RNA sequencing data |
| Other | ProteomeXchange Consortium | | Dataset Identifier: PXD037230 | Proteomics data |

## Mice

All animal studies were approved by the University Committee for Animal Resources (UCAR) and in compliance with ARRIVE guidelines. Scx-Cre mice were generously provided by Dr. Ronen Schweitzer. ROSA26-iDTR$^{F/F}$ (#007900) mice were obtained from the Jackson Laboratory (Bar Harbor, ME, USA). Scx-Cre mice were crossed to ROSA26-iDiphtheria Toxin Receptor$^{Lox-STOP-Lox}$ (DTR$^{F/F}$) mice to generate a model of Scleraxis-lineage tendon cell depletion (Scx-Cre$^+$; DTR$^{F/+}$; referred to as DTR). Expression of DTR is inhibited prior to Cre-mediated recombination due to the presence of a STOP cassette flanked by loxP sites (Lox-STOP-Lox). After Cre-mediated recombination, the STOP cassette is deleted, resulting in expression of DTR, specifically in Scleraxis-lineage cells. Thus, administration of diphtheria toxin (DT) in these mice results in apoptosis of Scleraxis-lineage cells. Scx-Cre$^-$; DTR $^{F/+}$ (WT) littermates were used as controls. DT was administered for 5 consecutive days in the hind paws of both DTR and WT mice at three months of age. Male and female mice were used for all studies. C57BL/6 J mice (B6; #664, Jackson Laboratory) were used for natural aging studies as noted. All mouse work (injections, surgeries, harvests) were performed in the morning. Mice were kept in a 12 hr light/dark cycle.

## Paraffin histology and immunofluorescence

Sections were stained with DAPI to visualize nuclei and imaged using a VS120 Virtual Slide Microscope (Olympus, Waltham, MA). Using ImageJ (*Schneider et al., 2012*), a region of interest (ROI) was drawn at the tendon midsubstance and the area of the ROI was calculated. Nuclei within the ROI were manually counted, and total nuclei number was normalized to area. An n=5–7 mice per group were used for quantification. Five-micron paraffin sagittal sections of 6 M WT, 6 M DTR, and 31 M C57BL/6 J hind paws were cut. For immunofluorescence, sections were stained with an anti-GBP2 antibody (1:500, Proteintech Cat#: 11854–1-AP, RRID: AB_2109336), followed by a Donkey anti-rabbit Rhodamine-Red-X secondary antibody (1:200, Jackson ImmunoResearch, #711-296-152, RRID: AB_2340614). Sections were counterstained with DAPI to visualize nuclei and imaged using a VS120 Virtual Slide Microscope (Olympus, Waltham, MA). GBP2 + cells and total nuclei within the ROI were manually counted and the number of GBP2 + cells was normalized by the total nuclei number to quantify the percent of GBP2 + cells. All quantification was conducted in a blinded manner. An n=6–8 mice per group were used for quantification.

## Second harmonic generation two-photon confocal imaging

Five-micron paraffin sections of DTR, WT, and C57BL/6 J hind paws were utilized for second harmonic generation (SHG) imaging. Sections were scanned with a Spectra-Physics MaiTai HP DeepSee Ti:Sapphire Laser, tuned to 1000 nm, under ×25 magnification, 2.5 X optical zoom, and step size of 0.25 mm. 3D projections of image stacks were generated using the 3D-Project macro in ImageJ and analyzed for collagen fibril uniformity using the built-in Directionality macro function. The Directionality (dispersion) macro utilizes Fourier transform analysis to derive spatial orientation of image stacks (*Lobo et al., 2015*). Sections were analyzed from n=3–5 mice per group per age.

## Quantification of biomechanical properties

DTR and WT and C57BL/6 J FDL tendons were harvested from the hind paws. Specifically, each FDL tendon was carefully separated at the myotendinous junction under a dissecting microscope. The tarsal tunnel was then cut and the FDL tendon was slowly released from the tarsal tunnel, isolated

until the bifurcation of the digits and then cut and released. Under the dissecting microscope, any additional connective tissues (e.g., muscle) were removed and the FDL tendon was prepared for uniaxial testing. Two pieces of sandpaper were placed on each end of the tendon and glued together using cyanoacrylate (Superglue, LOCTITE). All the steps above were performed with the tissue periodically submerged in PBS to avoid any potential tissue drying. Each gripped tendon was transferred into a semi-customized uniaxial microtester (eXpert 4000 MicroTester, ADMET, Inc, Norwood MA). The microtester, along with the sample, was transferred to an inverted microscope (Olympus BX51, Olympus) to visualize the tendon and quantify the gauge length, width, and thickness. The gauge length of each sample was set as the end-to-end distance between opposing sandpaper edges and was set the same for all samples tested. The cross-section of the tendon was assumed to be an ellipse, where the width of the tissue represents the major axis and the thickness of the tissue represents the minor axis. Based on the optically measured width and thickness of each tendon, the area of the elliptical cross-section was computed. After measurement of the tendon's gauge length, a uniaxial displacement-controlled stretching of 1% strain rate until failure was applied. Load and grip-grip displacement data were recorded and converted to stress-strain data, and the failure mode was tracked for each mechanically tested sample. The load-displacement and stress-strain data were plotted and analyzed to determine structural (*stiffness)* and material (*modulus*) properties. Specifically, the slope of the linear region from the load displacement graph was determined to be the stiffness of the tested sample. The slope of the linear region from the stress-strain graph was taken to equal the elastic modulus parameter of each tested tendon. Note that this calculation assumes that stress and strain are uniform within each specimen. A sample size of n=7 animals per group was utilized for biomechanical testing and analysis.

## Sample preparation for mass spectrometry

Homogenization of each tendon tissue was performed by adding 150 µL of 5% Sodium Dodecyl Sulphate (SDS), 100 mM Triethylammonium bicarbonate (TEAB). Samples were vortexed and then sonicated (QSonica) for 5 cycles, with a 1 min resting period on ice after each cycle. Lysates were then centrifuged at 15,000 x g for 5 min to collect cellular debris, and the supernatant was collected. Next, the total protein concentration was determined by bicinchoninic acid assay (BCA; Thermo Scientific), after which samples were diluted to 1 mg/mL in 5% SDS, 50 mM TEAB. A mass of 25 µg of protein from each sample was reduced with dithiothreitol to 2 mM, followed by incubation at 55 °C for 60 min. Iodoacetamide was added to 10 mM and incubated for 30 min in the dark at room temperature to alkylate the proteins. Phosphoric acid was added to 1.2%, followed by six volumes of 90% methanol, 100 mM TEAB. The resulting solution was added to S-Trap micros (Protifi) and centrifuged at 4000 x g for 1 min. S-Traps with the trapped protein were washed twice by centrifuging through 90% methanol, 100 mM TEAB. One µg of trypsin was brought up in 20 µL of 100 mM TEAB and added to the S-Trap, followed by 20 µL of TEAB to ensure the sample did not dry out. The cap to the S-Trap was loosely screwed on but not tightened to ensure the solution was not pushed out of the S-Trap during digestion. Samples were placed in a humidity chamber at 37 °C overnight. The next morning, the S-Trap was centrifuged at 4000 x g for 1 min to collect the digested peptides. Sequential additions of 0.1% Trifluoroacetic acid (TFA) in acetonitrile and 0.1% TFA in 50% acetonitrile were added to the S-trap, centrifuged, and pooled. Samples were frozen and dried down in a Speed Vac (Labconco), then re-suspended in 0.1% trifluoroacetic acid prior to analysis. Three tendons per genotype per timepoint were used for proteomic analysis.

## Mass spectrometry (MS)

Peptides were injected onto a homemade 30 cm C18 column with 1.8 µm beads (Sepax), with an Easy nLC-1200 HPLC (Thermo Fisher), connected to a Fusion Lumos Tribrid mass spectrometer (Thermo Fisher). Solvent A was 0.1% formic acid in water, while solvent B was 0.1% formic acid in 80% acetonitrile. Ions were introduced to the MS using a Nanospray Flex source operating at 2 kV. The gradient began at 3% B and held for 2 min, increased to 10% B over 6 min, increased to 38% B over 95 min, then ramped up to 90% B in 5 min and was held for 3 min, before returning to starting conditions in 2 min and re-equilibrating for 7 min, for a total run time of 120 min. The Fusion Lumos was operated in data-dependent mode, with MS1 scans acquired in the Orbitrap, and MS2 scans acquired in the ion trap. The cycle time was set to 2 s. Monoisotopic Precursor Selection (MIPS) was set to Peptide. The

full scan was done over a range of 375–1400 m/z, with a resolution of 120,000 at m/z of 200, an AGC target of 4e5, and a maximum injection time of 50 ms. Peptides with a charge state between 2 and 5 were picked for fragmentation. Precursor ions were fragmented by collision-induced dissociation (CID) using a collision energy of 30% with an isolation width of 1.1 m/z. The Ion Trap Scan Rate was set to Rapid, with a maximum injection time of 35ms, an AGC target of 1e4. Dynamic exclusion was set to 45 s.

### MS data filtering

Raw data was searched using the SEQUEST search engine within the Proteome Discoverer software platform, version 2.4 (Thermo Fisher), using the SwissProt *Mus musculus* database. Trypsin was selected as the enzyme allowing up to 2 missed cleavages, with an MS1 mass tolerance of 10 ppm, and an MS2 mass tolerance of 0.6 Da. Carbamidomethyl was set as a fixed modification, while oxidation of methionine was set as a variable modification. The Minora node was used to determine relative protein abundance between samples using the default settings. Percolator was used as the FDR calculator, filtering out peptides which had a q-value greater than 0.01.

### Filtered MS data analysis

To identify proteins that were significantly different between two groups, the log2 fold change (FC) and the -log10 of the p-value were plotted in a volcano plot and we utilized the most stringent cut-off parameters of log2FC >1 and -log10(p-value)>1.3. Significantly decreased proteins among two groups were inserted in PANTHER classification system (*Mi et al., 2021*) (http://pantherdb.org/) to classify protein type. To understand what types of biological processes (BPs), molecular functions (MFs), and cellular components (CCs) were shifted on each condition, we inserted all significantly decreased proteins in the gene ontology classification tool DAVID (*Huang et al., 2009*) (https://david.ncifcrf.gov/). Protein network analysis was performed using the Search Tool for Retrieval of Interacting Genes/Proteins (STRING), v11.0 (*Szklarczyk et al., 2019*), and ECM-related proteins were further classified using MatrisomeDB (*Hynes and Naba, 2012*) (https://web.mit.edu/hyneslab/matrisome/).

### Single-cell isolation

A total of 16 FDL tendons per group per age were pooled together into low glucose Dulbecco's Modified Eagle Medium (DMEM) and subsequently digested in Collagenase Type I (5 mg/ml) (Worthington Biochemical, Lakewood, NJ, LS004196) and Collagenase Type IV (1 mg/ml) (Worthington Biochemical, Lakewood, NJ, LS004188). Young WT tendons were completely digested after two hours, while young DTR and aged C57BL/6 J WT tendons after ninety minutes. After digestion, the single-cell suspension was filtered for any potential debris via a 70 µm and a 50 µm cell strainer and resuspended in Dulbecco's Phosphate Buffer Solution (dPBS). Next, the single cell suspension was centrifuged for 10 min at 500xg while being at 4 °C. After removal of supernatant, the single cell suspension was resuspended in 0.5% Bovine Serum Albumin (BSA).

### Single-cell RNA sequencing and analysis

Single-cell RNA sequencing (scRNAseq) was performed in collaboration with the UR Genomics Research Center. Libraries were prepared using a Chromium Single Cell 3 Reagent Kit (version 3, 10 X Genomics, Pleasanton, CA) following the directions of the manufacturer. Cells were loaded into the kit and processed using a chromium controller (10 X Genomics). Following library preparation and quality control (QC), libraries were sequenced using the S2 NovaSeq flow cell system (Illumina, San Diego, CA), generating an average of 65,000 reads per cell for all groups. Raw data quality control and downstream analyses were performed with R-4.1.2 and the Seurat package (4.0.6) (*Stuart et al., 2019*). Cells with <1000 or>5000 genes, as well as cells with >10% mitochondrial genes were removed from the dataset. After filtering out low quality cells, there were 8359 cells for the young (6 M old) WT group, 6554 cells for the young (6 M old) DTR group, and 4798 cells for the 21 M old C57BL/6 J WT group. All three different datasets were integrated to determine alterations of different cell subpopulations between groups using the 'FindIntegrationAnchors' function (dims = 1:20). The function 'IntegrateData' was applied in the anchor set with the default additional arguments. Next, data was scaled using the 'ScaleData' with default parameters and principal component analysis (PCA) was performed on the integrated dataset to compute 20 principal components (PCs). Uniform Manifold Approximation

and Projection (UMAP) dimensionality reduction was applied, and the Shared Nearest Neighbor (SSN) graph was constructed by utilizing dimensions 1:20 as input feature. To identify cell clusters, the 'FindClusters' function was utilized on the integrated dataset at a resolution of 0.1, resulting in seven distinct cell clusters. To identify genetic markers of each single cluster and annotate the different clusters to known cell types, the top differentially expressed genes of each cluster were identified in the heatmap (*Figure 3—figure supplement 1*). Next, the 'FindAllMarkers' function was used to identify upregulated genes for each cluster. The average gene expression level was calculated across each cluster. The minimum percentage of cells in which the gene is detected in each cluster was set to 50%. The average log2 change threshold was set to at least 0.25. Marker gene lists were then compared to a set of literature-defined gene markers via direct literature search and by utilizing *CellMarker* software (http://bio-bigdata.hrbmu.edu.cn/CellMarker/search.jsp) (*Zhang et al., 2019*). Significance was determined using Wilcoxon rank sum test with p values adjusted based on Bonferroni correction applying all features in the data set (p_val_adj <0.05). The tenocytes cluster from the integrated data was subset out from the rest using the 'Subset' and were independently re-clustered following the same steps described above.

## Cell-cell communication analysis

Cellular autocrine and paracrine communication was performed using the R *CellChat* package version 1.4.0 (*Jin et al., 2021*; *Jin et al., 2022*). To assess tenocyte-tenocyte communication, tenocyte clusters 1–3 from the integrated data were subsetted and a CellChat object of tenocytes 1–3 was created for each condition using the 'createCellChat' function. After proper annotation, the 'CellChatDB.mouse' database was utilized for our study To identify overexpressed genes, the 'identifyOverExpressedGenes' function was utilized. To identify significant cellular interactions, the 'identifyOverExpressedInteractions' function was utilized. Finally, to infer communication probabilities, the 'computeCommunProb' function was utilized. Next, to study the effect of depletion and natural aging on tenocyte-tenocyte communication, a merged CellChat object was created for the 6 M WT vs 6 M DTR, and 6 M WT vs 21 M B6 groups, respectively, using the 'mergeCellChat' function. To assess tenocyte-macrophage communication, tenocytes and macrophage clusters from the integrated data were subsetted. The same steps mentioned above for the tenocyte-tenocyte communication analysis were also followed for tenocyte-macrophage communication. To identify the differential number of interactions between two conditions, the 'netVisual_diffInteraction' function was utilized. To identify specific signaling pathways that were conserved or expressed only in one condition, the 'rankNet' function was utilizes. Next, the 'netAnalysis_signalingRole_heatmap' function were applied to identify the 'outgoing' and 'incoming' patterns of signaling pathways for each different cell population and how that compares between 2 conditions. Finally, specific ligand-receptor interactions were identified for specific signaling pathways of interest utilizing the 'netVisual_bubble' function.

## Statistical analysis

Sample sizes for cell density, dispersion, and biomechanical properties were determined based on post-hoc power calculations of previously published work (*Best et al., 2021*; *Ackerman et al., 2019*). Quantitative data was analyzed via GraphPad Prism and is presented as mean ± standard deviation (SD). Either a student's t-test or two-way analysis of variance (ANOVA) with Tukey's test was used as appropriate. Mice were randomly selected for specific experimental outcome metrics prior to the start of an experiment or surgery and quantitative data (e.g. DAPI quantification, SHG dispersion levels, biomechanical properties) were analyzed in a blinded manner. For all experiments, an n=1 represents one mouse. p-values ≤0.05 were considered significant. * Indicates $p<0.05$, ** indicates $p<0.01$, *** indicates $p<0.001$, **** indicates $p<0.0001$ (*Perez-Riverol et al., 2022*).

## Acknowledgements

This work was supported in part by NIH/ NIAMS R01AR073169 and R01AR077527 (to AEL), and K99 AR080757 (to AECN). The Histology, Biochemistry and Molecular Imaging (HBMI) Core was supported by NIH/ NIAMS P30AR069655. The content is solely the responsibility of the authors and does not necessarily represent the official views of the National Institutes of Health.

## Additional information

### Funding

| Funder | Grant reference number | Author |
|---|---|---|
| National Institute of Arthritis and Musculoskeletal and Skin Diseases | R01AR073169 | Alayna E Loiselle |
| National Institute of Arthritis and Musculoskeletal and Skin Diseases | R01AR077527 | Alayna E Loiselle |
| National Institute of Arthritis and Musculoskeletal and Skin Diseases | K99 AR080757 | Anne EC Nichols |

The funders had no role in study design, data collection and interpretation, or the decision to submit the work for publication.

### Author contributions

Antonion Korcari, Conceptualization, Formal analysis, Investigation, Writing – original draft; Anne EC Nichols, Conceptualization, Formal analysis, Methodology, Writing - review and editing; Mark R Buckley, Resources, Methodology, Writing - review and editing; Alayna E Loiselle, Conceptualization, Funding acquisition, Writing - review and editing

### Author ORCIDs

Anne EC Nichols ⓘ http://orcid.org/0000-0001-8754-7735
Alayna E Loiselle ⓘ http://orcid.org/0000-0002-7548-6653

### Ethics

This study was performed in strict accordance with the recommendations in the Guide for the Care and Use of Laboratory Animals of the National Institutes of Health. All animal studies were approved by the University Committee for Animal Resources (UCAR) (protocol 2014-004E).

### Decision letter and Author response

Decision letter https://doi.org/10.7554/eLife.84194.sa1
Author response https://doi.org/10.7554/eLife.84194.sa2

## Additional files

### Supplementary files
• MDAR checklist

### Data availability

Single cell RNA sequencing data has been deposited at Gene Expression Omnibus (GEO) (Accession # GSE214929) and are publicly available as of the date of publication. The mass spectrometry proteomics data have been deposited to the ProteomeXchange Consortium via the PRIDE partner repository (*Perez-Riverol et al., 2022*) with the dataset identifier PXD037230.

The following previously published datasets were used:

| Author(s) | Year | Dataset title | Dataset URL | Database and Identifier |
|---|---|---|---|---|
| Korcari A, Nichols AEC, Buckley MR, Loiselle AE | 2023 | Scleraxis-lineage cells are required for tendon homeostasis and their depletion induces an accelerated extracellular matrix aging phenotype | https://www.ncbi.nlm.nih.gov/geo/query/acc.cgi?acc=GSE214929 | NCBI Gene Expression Omnibus, GSE214929 |
| Korcari A, Nichols AEC, Buckley MR, Loiselle AE | 2023 | Scleraxis-lineage cells are required for tendon homeostasis and their depletion induces an accelerated extracellular matrix aging phenotype | https://www.ebi.ac.uk/pride/archive/projects/PXD037230 | PRIDE, PXD037230 |

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
