## [Editor Report]

This fundamental work advances our understanding of the cellular and molecular changes of the aged tendon. The evidence supporting the conclusion is convincing, using a DTR-based ScxLin cell depletion model along with state-of-art proteomic and scRNA-seq analyses. This paper is of potential interest to scientists and physicians who study the mechanisms of the tendon aging process.

---

## [Decision Letter]

**Decision letter after peer review:**

Thank you for submitting your article "Scleraxis-lineage cells are required for tendon homeostasis and their depletion induces an accelerated extracellular matrix aging phenotype" for consideration by *eLife*. Your article has been reviewed by 2 peer reviewers, and the evaluation has been overseen by a Reviewing Editor and Carlos Isales as the Senior Editor. The following individual involved in review of your submission has agreed to reveal their identity: Nathanial Dyment (Reviewer #1).

Essential revisions:

This is a generally well-executed set of study, the data from which largely supports the conclusions. The manuscript would be strengthened by (1) improved clarity of the methods and figures as pointed out by the reviewers; (2) including more introduction/discussion on Gbp2; and (3) including more discussion on the limitations of using the DTA mice to model the aging process (eg. artificial, not a perfect model of aging…).

*Reviewer #1 (Recommendations for the authors):*

This study developed a novel model of accelerated tendon extracellular matrix (ECM) aging via depletion of Scleraxis-lineage (ScxLin) cells in young mice (DTR). The authors found the depletion reduced cell numbers to similar baselines as aged tendons, indicating that a minimum cell number threshold exists to maintain tendon. This cell loss coincided with disrupted ECM organization and reduced mechanical properties. The DTR and aged tendons had similar protein composition with the main difference compared to young healthy tendons being a loss of high turnover ECM proteins. Via sc-seq, DTR and aged tendon had fewer biosynthetic cells, correlating with loss of certain ECM proteins. Interestingly, the remaining cells in the DTR model differed from aged tendons. While somewhat artificial, this depletion model system is an interesting way to investigate mechanisms that lead to reduced ECM turnover and matrix degeneration. I commend the authors for conducting this extensive study and have mostly minor comments to help with data interpretation and presentation.

1. In Figure 3E, the authors show the relative proportions of tenocytes in the 6M WT, 6M DTR, and 21M B6 groups, what about other cell types? It would be great to show pie charts of all of the cell clusters in one of the supplemental figures (maybe Figure S4). Did the other cell populations increased or decrease? I know that the authors plan to publish the epitenon data in a separate paper but it would still help the readers in this paper to understand the relative proportions.

2. In Figure 4F, it appears that there is a subset within the Tenocyte 1 cluster (right 1/3 of the heatmap for Tenocyte 1). Is this a unique subset? Did the authors conduct upregulated biological processes on this subset?

3. The authors used GBP2 as a unique marker for the tenocyte 1 population but didn't discuss it further. What is the function of GBP2 and does it help explain some of the degenerative changes seen in aging tendons?

4. Similar to my previous point, Thbs signaling was demonstrated to be a common pathway in both DTR and aging. What do we know about this signaling axis and its role in tendon? It would be great to touch on Thbs and GBP2 in the discussion.

5. Can the authors please clarify the following sentence in the discussion? Which data in the paper support this statement? "Finally, these data suggest that the initiation and early evidence of degeneration occurs much earlier in life than previously appreciated."

6. Reference 1 and 36 are the same.

7. The fact that the cell density 3M after DTR reached the same minimum threshold as aged tendons and that DTR tendons didn't progressively lose more cells with age is quite interesting, as the authors indicated.

*Reviewer #2 (Recommendations for the authors):*

In this manuscript, the authors characterize the proteomic and single RNA Seq profiles of tendons that are aged or depleted of Scx+ cells. They show that the tendons depleted of Scx+ cells share a similar cellularity, collagen disorganization, and impaired biomechanics as the aged tendon. They find that ECM proteins/regulators are mis-expressed in Scx depleted and aged tendons, with the aged tendons having a population of tenocytes that expression a program of inflammatory markers.

1) In all figures, font on the heat maps are out of focus and very small – would make it easier for the reader if this was improved.

2) Methods – For the biomechanical testing – it is not clear how was the strain applied to the tendon controlled to be the same for each tendon.

3) Methods – For the DT toxin injections, what time points were the mice injected?

4) Figure 1D- there may be a labeling error of the graph – 3M post DT is not the same as 6M old WT.

5) With the proteomics data, how do you know if the decrease in protein expression of these ECM molecules is due to the decrease in cellularity or each tendon cell has decreased expression of these markers? Would be useful to validate this with expression of a candidate protein marker in WT /DT treated/aged tenocytes.

6) Please define Gbp2 – what is this molecule?

7) In Figure 5 – there is graphic representation of genes enhanced in the different mouse groups, but are there any genes downregulated compared to WT?

8) In the discussion, the authors mention that it would be helpful to prevent apoptosis of tenocytes during aging – where is the evidence or citation that shows cell death is increased in aged tenocytes?

---

## [Author Response]

Reviewer #1 (Recommendations for the authors):This study developed a novel model of accelerated tendon extracellular matrix (ECM) aging via depletion of Scleraxis-lineage (ScxLin) cells in young mice (DTR). The authors found the depletion reduced cell numbers to similar baselines as aged tendons, indicating that a minimum cell number threshold exists to maintain tendon. This cell loss coincided with disrupted ECM organization and reduced mechanical properties. The DTR and aged tendons had similar protein composition with the main difference compared to young healthy tendons being a loss of high turnover ECM proteins. Via sc-seq, DTR and aged tendon had fewer biosynthetic cells, correlating with loss of certain ECM proteins. Interestingly, the remaining cells in the DTR model differed from aged tendons. While somewhat artificial, this depletion model system is an interesting way to investigate mechanisms that lead to reduced ECM turnover and matrix degeneration. I commend the authors for conducting this extensive study and have mostly minor comments to help with data interpretation and presentation.1. In Figure 3E, the authors show the relative proportions of tenocytes in the 6M WT, 6M DTR, and 21M B6 groups, what about other cell types? It would be great to show pie charts of all of the cell clusters in one of the supplemental figures (maybe Figure S4). Did the other cell populations increased or decrease? I know that the authors plan to publish the epitenon data in a separate paper but it would still help the readers in this paper to understand the relative proportions.

Thank you for your comment, we now include new Figure 3—figure supplement 1D, which has a bar chart depicting the % of each cell cluster across the three conditions (6M WT, 6M DTR, 21M WT). Key take homes from these data include the high proportion of epitenon cells that are captured, including 58% of cells in the 21M WT group. While we speculate this may be due to combination of the low cellularity in the aged tendons, as well as the superficial nature of the epitenon, which may make these cells more amenable to digest. However, further work is needed to validate this speculation. Furthermore, there was a greater proportion of tendon resident macrophages captured in the depleted and aged groups, suggesting these populations may be more resistant to age-related declines.

2. In Figure 4F, it appears that there is a subset within the Tenocyte 1 cluster (right 1/3 of the heatmap for Tenocyte 1). Is this a unique subset? Did the authors conduct upregulated biological processes on this subset?

This is a great point, indeed the right 1/3 of the heatmap for tenocytes 1 in Figure 4F exhibits some DEGs compared to the rest 2/3 of that same cluster, with this segment representing part of the Tenocytes 1 cluster that is only found in aged tendons. As we discuss in Figure 5F-J, these aged cells demonstrate up-regulation of age-related and pro-inflammatory genes such as IL6, Hspa1a, Hspa1b, Mt1, Mt2 and the downregulation of genes related to ECM/collagen biosynthesis and ECM organization such as Col6a3, Col1a1, Sparc, Vcan, Chad, Col1a2, and thus demonstrate shifts in biological processes.

3. The authors used GBP2 as a unique marker for the tenocyte 1 population but didn't discuss it further. What is the function of GBP2 and does it help explain some of the degenerative changes seen in aging tendons?

Thank you for this suggestion. Based on this, we now provide additional text related to GBP2. While GBP2 has not been studied in the context of tendon, or musculoskeletal biology, GBP2 is downstream of IFN-γ signaling and exhibits host defense and anti-microbial functions [Tretina et al., J. Exp Med 2019; Kutsch et al., FEBS J 2021]. Consistent with this, Tenocytes 1, which are defined by specific expression of GBP2, demonstrate biological processes (BPs) related to innate immune and defense response such as response to cytokine, innate immune response, response to IL-1 (Figure 4F, G). We have clarified that expression of GBP2 is an effective marker of Tenocytes 1, but the functional role of GBP2 in tenocytes remains to be determined.

4. Similar to my previous point, Thbs signaling was demonstrated to be a common pathway in both DTR and aging. What do we know about this signaling axis and its role in tendon? It would be great to touch on Thbs and GBP2 in the discussion.

Thank you for this suggestion. We have added additional text to the discussion to address this. Syndecans are one of the main transmembrane proteins that bind with (thrombospondins) TSPs and since there was a decrease of TSPs in both DTR and aging, that resulted in a similar loss of TSP-Syndecan pathway among the two conditions. TSP2 -Syndecan 4 pathway exhibited the highest ligand-receptor communication in terms of statistical significance and communication probability in the young WT, and it was completely lost with both DTR and aging.

While there are no studies focusing on the roles of Syndecans in the tendon field, there is a limited number of studies regarding the roles of Syndecans in musculoskeletal tissues. Cornelison et al., as well as Ronning et al., have shown that Syndecan-4 KO mice exhibit impairments muscle development, homeostasis, and healing after injury [Cornelison et al., Genes Dev 2004; Ronning et al., Front Cell Dev Biol 2020].

The text now states: “Finally, our CellChat analyses identified that Thbs2-Sdcn4 signaling was significantly impaired with both DTR and aging, suggesting that this pathway might be important for maintenance of tendon homeostasis. While this specific pathway has not been extensively studied in tendon, previous work has shown that Sdcn 4 is required for muscle development, homeostasis, and healing after injury ^82, 83^. As such, future studies can better define the role of this signaling axis in tendon health, as it may represent promising therapeutic target.”

5. Can the authors please clarify the following sentence in the discussion? Which data in the paper support this statement? "Finally, these data suggest that the initiation and early evidence of degeneration occurs much earlier in life than previously appreciated."

This sentence has been clarified. Our intention was to highlight the greater understanding of the timeline of age-related tendon degeneration that our data provide. More specifically, since we show that the decline in cell density occurs at 10-12M of age (approximately middle age), and subsequently initiates a degenerative cascade, including loss of high-turnover rate ECM components, these data provide greater temporal context on the age-related degeneration phenotypes identified in prior studies. For example, many prior studies have identified substantial declines in tendon cell density with advanced aging (e.g., 21-24M), relative to young mice. However, based on these studies it was unknown when precisely the drop in tendon cell density occurred, or even whether this was related to age-related degeneration.

The revised text now states: “Finally, these data provide more granular information on the timeline in which age-associated declines in tendon cell density actually occur and provide a functional and mechanistic link between declines in tendon cell density and initiation of tendon degeneration.”

6. Reference 1 and 36 are the same.

Thanks for pointing this out, we have removed reference 36.

7. The fact that the cell density 3M after DTR reached the same minimum threshold as aged tendons and that DTR tendons didn't progressively lose more cells with age is quite interesting, as the authors indicated.

Thank you for this comment. As the reviewer notes, there seems to be a minimum threshold for cell density, and the mechanisms that regulate this threshold representing an intriguing mechanism that we will pursue in the future. For example, would a higher dose of DT, or additional doses of DT be able to force cell density below this threshold?

Reviewer #2 (Recommendations for the authors):In this manuscript, the authors characterize the proteomic and single RNA Seq profiles of tendons that are aged or depleted of Scx+ cells. They show that the tendons depleted of Scx+ cells share a similar cellularity, collagen disorganization, and impaired biomechanics as the aged tendon. They find that ECM proteins/regulators are mis-expressed in Scx depleted and aged tendons, with the aged tendons having a population of tenocytes that expression a program of inflammatory markers.1) In all figures, font on the heat maps are out of focus and very small – would make it easier for the reader if this was improved.

We apologize for this. We have updated these figures with higher resolution and larger font.

2) Methods – For the biomechanical testing – it is not clear how was the strain applied to the tendon controlled to be the same for each tendon.

This is a great point. To clarify, after prepping the tendons for biomechanical testing, we measured the gauge length of each tendon and applied a displacement-controlled stretching of 1% of the initial gauge length per second until failure (1% strain rate). The gauge length was kept consistent between all groups, thus the strain rate applied to all tendons was consistent. Moreover, we have previously validated the experimental measurements of the tendon biomechanical properties using Finite Elements Analysis (Korcari et al., JMBBM, 2022).

3) Methods – For the DT toxin injections, what time points were the mice injected?

Thank you for your comment, as shown in Figure 1A, 3M old mice (both WT and DTR) were injected with DT for 5 consecutive days and then harvested after 3, 6, and 9M post depletion. To increase clarity in the manuscript, we have also included this info in the methods Mice section.

4) Figure 1D- there may be a labeling error of the graph – 3M post DT is not the same as 6M old WT.

Thank you for identifying this issue. To clarify- both WT (Cre-) and DTR (Cre+) mice were injected with DT at 3M of age. The 3M post-DT cohort was thus harvested at 6M of age. In Figure 1D, data from both the 6M old WT (3M post-DT) and the 6M old DTR (3M post-DT) groups are included as comparisons lines. We have added additional information to Figure 1A to make the experimental timeline and relative ages more clear.

5) With the proteomics data, how do you know if the decrease in protein expression of these ECM molecules is due to the decrease in cellularity or each tendon cell has decreased expression of these markers? Would be useful to validate this with expression of a candidate protein marker in WT /DT treated/aged tenocytes.

This is an important point. To address this, we used Chad and Coch as example. These ECM proteins were significantly decreased in aged and DTR tendons in the proteomics analysis (Figure 2M). We then examined these genes in the single cell RNA seq dataset (Figure 2—figure supplement 3). At the transcript level, Tenocytes 1 and Tenocytes 3 are the predominant populations expressing Coch and Chad. While the cells that remained in the 6M DTR and the 21M B6 group demonstrated high levels of Coch and Chad expression (Figure 2—figure supplement 3A and D), there was a substantial decline in the total number and the proportion of CocH^+^ and Chad+ cells in these groups (Figure 2—figure supplement 3C and F). To further confirm the data in Figure 2—figure supplement 3A and D, we have plotted the mRNA expression levels of *Coch*, *Chad* and other ECM components that are down-regulated with DTR and/or aging in Author response image 1. Collectively, these data suggest that it is a decline in the number of cells that express these ECM components rather than down-regulation of expression on a per-cell basis.

**Author response image 1. sa2fig1:** 

6) Please define Gbp2 – what is this molecule?

Thank you for this suggestion. Based on this, we now provide additional text related to GBP2. While GBP2 has not been studied in the context of tendon, or musculoskeletal biology, GBP2 is downstream of IFN-γ signaling and exhibits host defense and anti-microbial functions [Tretina et al., J. Exp Med 2019; Kutsch et al., FEBS J 2021]. Consistent with this, Tenocytes 1, which are defined by specific expression of GBP2, demonstrate biological processes (BPs) related to innate immune and defense response such as response to cytokine, innate immune response, response to IL-1 (Figure 4F, G). We have clarified that expression of GBP2 is an effective marker of Tenocytes 1, but the functional role of GBP2 in tenocytes remains to be determined.

7) In Figure 5 – there is graphic representation of genes enhanced in the different mouse groups, but are there any genes downregulated compared to WT?

Thank you for this question. As shown in Figure 5 there are many differentially expressed (significantly increased or decreased) genes, across each set of comparisons. Using Figure 5B as an example, we compare differential gene expression in 6M WT Tenocytes 1 vs. 6M DTR Tenocytes 1. Genes in blue are significantly increased in 6M WT Tenocytes 1 vs. 6M DTR Tenocytes 1, that is, these are genes that are downregulated in the 6M DTR Tenocyte 1 cluster. In contrast, genes that are significantly reduced in 6M WT Tenoctyes 1 vs. 6M DTR Tenocytes (increased expression in 6M DTR Tenocyte 1) are shown in red. In addition, the heatmaps in Figure 5C, H, and M show up-regulation (purple) and down-regulation (yellow) of genes in 6M WT, relative to their respective comparison (Figure 5C: 6M DTR Tenocytes 1, Figure 5H: 21M B6 Tenocytes 1, Figure 5M: 6M DTR Tenocytes 2)

8) In the discussion, the authors mention that it would be helpful to prevent apoptosis of tenocytes during aging – where is the evidence or citation that shows cell death is increased in aged tenocytes?

Thank you for pointing this out. While we do have preliminary data supporting an age-dependent degree of apoptosis (increased cleaved caspase 3 at 9-10 months, followed by a decline at 12-13 months), we have not included these data in this manuscript, and have therefore revised this statement. The revised text now states: “As such, we have identified preventing the loss of biosynthetic tenocytes (whether via apoptosis or other mechanisms) as a critical intervention point for maintaining tendon structure-function.”